# Exploring the Benefits of Phycocyanin: From Spirulina Cultivation to Its Widespread Applications

**DOI:** 10.3390/ph16040592

**Published:** 2023-04-14

**Authors:** Raquel Fernandes, Joana Campos, Mónica Serra, Javier Fidalgo, Hugo Almeida, Ana Casas, Duarte Toubarro, Ana I. R. N. A. Barros

**Affiliations:** 1Mesosystem, Rua da Igreja Velha 295, 4410-160 Vila Nova de Gaia, Portugal; 2UCIBIO (Research Unit on Applied Molecular Biosciences), REQUIMTE (Rede de Química e Tecnologia), MEDTECH (Medicines and Healthcare Products), Laboratory of Pharmaceutical Technology, Department of Drug Sciences, Faculty of Pharmacy, University of Porto, 4050-313 Porto, Portugal; 3CBA and Faculty of Sciences and Technology, University of Azores, Rua Mãe de Deus No 13, 9500-321 Ponta Delgada, Portugal; 4Centre for the Research and Technology of Agro-Environmental and Biological Sciences (CITAB), Institute for Innovation, Capacity Building and Sustainability of Agri-Food Production (Inov4Agro), University of Trás-os-Montes and Alto Douro (UTAD), Quinta de Prados, 5000-801 Vila Real, Portugal

**Keywords:** *Arthrospira*, Spirulina, phycocyanin, extraction methods, purification methods, applications, circular economy

## Abstract

Large-scale production of microalgae and their bioactive compounds has steadily increased in response to global demand for natural compounds. Spirulina, in particular, has been used due to its high nutritional value, especially its high protein content. Promising biological functions have been associated with Spirulina extracts, mainly related to its high value added blue pigment, phycocyanin. Phycocyanin is used in several industries such as food, cosmetics, and pharmaceuticals, which increases its market value. Due to the worldwide interest and the need to replace synthetic compounds with natural ones, efforts have been made to optimize large-scale production processes and maintain phycocyanin stability, which is a highly unstable protein. The aim of this review is to update the scientific knowledge on phycocyanin applications and to describe the reported production, extraction, and purification methods, including the main physical and chemical parameters that may affect the purity, recovery, and stability of phycocyanin. By implementing different techniques such as complete cell disruption, extraction at temperatures below 45 °C and a pH of 5.5–6.0, purification through ammonium sulfate, and filtration and chromatography, both the purity and stability of phycocyanin have been significantly improved. Moreover, the use of saccharides, crosslinkers, or natural polymers as preservatives has contributed to the increased market value of phycocyanin.

## 1. Introduction

The continued growth of the world’s population is becoming a global public health problem as it leads to food shortages. It is expected that the world population could reach up to 9.9 billion people by 2050 [1]. The current food industry cannot meet the demand for food, so alternative sources, such as natural foods, need to be explored [2]. In this context, marine resources, especially microalgae, are gaining global interest to meet this requirement due to their high nutritional profile as alternative sources of proteins, pigments, antioxidants, carbohydrates, and lipids, as well as the variety of high-value molecules that can be used in different industries [3,4] In addition, there is a need for new bioactive molecules due to increasing antibiotic resistance, which is expected to be the leading cause of death by 2050 [5]. The potential of microalgae-derived metabolites as natural alternatives for developing analogous compounds of antibiotics and other drugs to treat various diseases has been demonstrated, showing great promise for application in the pharmaceutical industry [6]. However, natural resources are often highly exploited to fuel global growth, often leading to depletion of these resources, degradation of soil quality, and increased burning of fossil fuels. To address these challenges, innovative solutions that prioritize the conservation and use of renewable resources are crucial, as well as policies that promote sustainable resource use and the development of clean technologies [7]. In this context, microalgae have been explored as an effective biological system that contributes to the growth economy by converting solar energy into organic compounds, including biofuels, dietary supplements, pigments, and antioxidants [8]. Due to all these factors, the demand for algae products is expected to increase significantly in the coming years. In fact, it has been estimated that the global algae market will reach a value of $6.3 billion by 2028, up from $4.5 billion in 2021 [9]. Of the many microalgae available, *Arthrospira platensis* (*A. platensis*), commonly known as Spirulina, is the most widely used natural resource, especially in the food industry. Furthermore, besides its exceptional nutritional properties, Spirulina is the most extensively researched microalgae [10] owing to its superior protein content. Spirulina cultivation is cost-effective and can be easily harvested, and it can thrive in diverse conditions such as high pH and heterotrophic or mixotrophic conditions, thus minimizing the risk of contamination by other microorganisms. It can be utilized as a dietary supplement for mammals and as a bioadsorbent for heavy metals. Its nutrients are highly bioavailable, enabling rapid absorption by the human body [11]. Additionally, Spirulina is rich in value-added compounds, mainly polyunsaturated fatty acids, and pigments, particularly phycocyanin. These properties make it a very valuable microalgae used in commercial and industrial contexts [10]. Phycocyanin is a water-soluble, non-toxic, and blue colored photosynthetic pigment reportedly used in food, cosmetic, and pharmaceutical industries [12]. Over the years, the biological functionality of phycocyanin has been extensively researched (e.g., antioxidation, inflammation, cancer, antimicrobial activity, neurodegeneration, diabetes, wound healing, and hyperpigmentation) [13]. However, the percentage of extraction of phycocyanin from Spirulina, the efficiency of the cell disruption method, and the instability of phycocyanin both under storage conditions and in product formulations limit its applicability. Optimized conditions for Spirulina culture to produce phycocyanin, extraction and purification conditions to achieve high purity, and the stability of phycocyanin for large-scale production are therefore sparking the interest of the scientific community [12]. Hence, the aim of this review is to update the main bio functionality and related applications of phycocyanin from its main source, Spirulina. In addition, the optimal Spirulina culture conditions for phycocyanin production, extraction and purification methods, and physical and chemical conditions affecting the purity and stability of this protein are described. The knowledge presented in this review can be used as a guide to the best conditions needed to improve phycocyanin production in different industries, reducing the associated costs and time, and increasing the stability and purity of phycocyanin according to the desired application.

## 2. Overview of Spirulina

*Arthrospira* is a well-known type of blue-green algae, frequently used as food supplement. It is a multicellular, filamentous, and photosynthetic cyanobacteria, whose life cycle comprises several stages, such as trichome fragmentation, hormogonia cell enlargement and maturation, and trichome elongation. The mature trichomes divide into filaments (2–16 μm) or hormogonia cells of different sizes, which reproduce by binary fission and take a helical or spiral shape [14,15]. Although *Arthrospira* and *Spirulina* are currently are two different genera, the term “Spirulina” is still commonly used to refer to two particular *Arthrospira* species, *Arthrospira maxima* and (the most common) *A. platensis* [15]. In nature, Spirulina is mainly found in saline alkaline lakes, particularly in subtropical and tropical areas, namely, Central and South America, Africa, and Asia [8].

Historically, Spirulina has been cultivated for centuries. In fact, it has been used as a primary food source since the 16th century by Aztec and Maya civilizations (Mexico) and Indians as Kanembous (Africa). Remarkably, it was not until the 1960s that Spirulina was rediscovered and studied by scientists regarding its biochemical composition and associated biological activities. Due to its high nutrient content, Spirulina was recognized as a “wonderful future food source” by the International Association of Applied Microbiology in 1967, and later by the United Nations World Food Conference in 1974 and then by the World Health Organization in 1996 as the “best food for the future” [16,17,18]. 

As any food or dietary supplement, the commercialized Spirulina is subject to regulations, such as requirements for the production process, including ensuring the absence of contaminants (e.g., microcystins, toxic metals, and pathogenic bacteria), labeling, and packaging, in order to ensure its safety and quality. In the United States (US), Spirulina is regulated by the US Food and Drug Administration (FDA, Maryland, United States of America), which has certified Spirulina as “Generally Recognized as Safe (GRAS)” (GRN 127) at dosages of 3–6 g *per* day [16,19,20]. In the European Union (EU, Brussels, Belgium), only the microalgae known as GRAS may be sold for human consumption, while other microalgae must be registered as novel foods [21]. Consequently, Spirulina was approved for commercialization by the European Commission without considering the Regulation (EU) 2015/2283. This regulation sets rules for placing novel foods (foods that have not been consumed by humans to a significant extent in the EU before May 1997) in the EU market [EU, 2015] [22]. Currently, Spirulina is commercialized in various forms, such as dry powder, capsules, and tablets, with the standard and recommended daily dosage, including those by manufacturers, being 0.25 g to 5 g [18,23,24]. However, a US National Library of Medicine report suggests a counseling session with a healthcare guide to determine the best dose for a specific condition [25]. Spirulina is, indeed, a nutrient-rich source, with a significant content of protein (55–70%) in its dry weight, and the essential amino acids (47%) (Table 1). Remarkably, the protein proportion of Spirulina is higher than commonly-used plant or animal protein sources, such as soybeans (35%), peanuts (25%), cereals (8–14%), meat and fish (15–25%), eggs (12%), milk powder (35%) and whole milk (3%) [16,18,26,27]. Regarding other nutrients, Spirulina is also rich in carbohydrates (15–25%), mainly polysaccharides (glucosans and rhamnosans) and mono- or disaccharides (glucose, fructose, sucrose). It also contains lipids (6–8%), of which a high amount are essential fatty acids (1.3–15%), predominantly palmitic acid, γ-linolenic acid, linoleic acid, and oleic acid. In addition, Spirulina contains all the essential minerals (7–13%; e.g., potassium, calcium, chromium, copper, iron, magnesium, manganese, phosphorus, selenium, sodium, and zinc), including relevant vitamins, particularly several B vitamins (B1, B2, B3, B6, B9, and B12), as well as vitamins C, D and E. Natural photosynthetic pigments are found in Spirulina, which is the main source of phycocyanin (14–20%), but chlorophylls (1%) and carotenoids (0.5%) can be also found [16,18,26,28] (Table 1).

Moreover, several health benefits of Spirulina have been associated with its biological activities, allowing its applications in different fields, such as the food, cosmetic, and pharmaceutical industries. In fact, scientific evidence has demonstrated several significant properties, such as probiotic, neuroprotective, hypoglycemic, hypolipidemic, anti-inflammatory, antioxidant, antimicrobial, anticancer, anti-nephrotoxic, anti-genotoxic, and anti-obesity activities [23,26,28,30,31]. 

Due to Spirulina’s impressive nutritional profile and associated health benefits, there is an increasing interest in the market. In fact, Spirulina is currently the leading microalgae species, namely *A. platensis*, with an annual production about 10,000 tons, mainly produced in China (~66% of total world production) [32]. Moreover, a market research report in collaboration with the European Algae Biomass Association (EABA, Florence, Italy) revealed that the global Spirulina market is expected to reach $968.6 million by 2028, at a Compound Annual Growth Rate (CAGR) of 13.2% between 2021 and 2028. Despite North America’s dominance in the Spirulina market, Asia-Pacific countries are expected to have a significant growth due to several factors, such as local and regional players, a growing dietary supplement industry, an increasing need to eradicate malnutrition, favorable climatic conditions, and the low cost of production [33]. 

Spirulina production systems can be operated either in open or closed loop, with the former being more common. In open-loop systems, Spirulina is cultivated in outdoor shallow tanks or ponds filled with water and nutrients. The tanks need to be continuously stirred to prevent Spirulina from sinking to the bottom and to ensure that it receives adequate light and nutrients. While in closed-loop systems, the tanks are completely closed and isolated from outside sources, such as a photobioreactor, which allows the precise control of the growing conditions. Regardless of the type and size of the system, namely, large-scale for commercial purposes or smaller scale for personal use, proper maintenance with regular cleaning and monitoring for possible contamination is important [10,27]. Additionally, it is crucial to ensure optimal growing conditions with precise temperature, light, and nutrient levels to produce high-quality Spirulina [8,34]. 

Currently, due to a greater demand of natural compounds, the large-scale cultivation of Spirulina is also focused on the production of high-value proteins, mainly phycocyanin (blue pigment), which, according to its purity grade, can be used in the food, cosmetic, and pharmaceutical industries [6,31,33]. In fact, the global phycocyanin market is expected to reach $245.5 million by 2027 [35]. The best conditions reported to cultivate Spirulina for phycocyanin production are a combination of temperature around 30 °C, light intensity of 300 μmol photon/m^2^/s, pH of 10.0–10.5, and medium containing fresh water, sodium bicarbonate, nitrates, phosphates, sulphates, and microelements. Although the nitrogen source is particularly important for phycocyanin production, other sources, such as urea, ammonium chloride, ammonium sulphate, and acid ammonium phosphate can also be used [12]. 

In addition, there has been an increased effort to optimize Spirulina production to improve economic feasibility and environmental sustainability. For example, producers are currently investing more in the valorization of residual biomass and the reuse and/or recycling of waste resources, which promotes a circular economy. Even after the phycocyanin is removed from Spirulina biomass, it still contains high amounts of antioxidants, vitamins, and minerals, making it a valuable raw material in skincare products and cosmetics. 

Recently, several sustainable microalgae biorefinery technologies have been studied to support circular economy. Particularly, in Spirulina cultivation, other metabolites have been explored through different biorefinery routes and used in other industries, such as human food and animal feed (as they contain useful nutrients such as carbohydrates, proteins, and vitamins), natural dyes (biopigments), biofuels production (from Spirulina biomass, enriched with lipids, proteins, and carbohydrates), biomaterials (e.g., polymer synthesis as polyhydroxyalkanoates and polyhydroxybutyrates), biofertilizers (residual Spirulina biomass), and wastewater treatment (microalgae as phycoremediation agent). Replacing the chemical nutrient medium with wastewater medium would also help reduce environmental impact and enable sustainable production on a commercial scale [2,23,24,36,37,38,39,40].

## 3. Phycocyanin

### 3.1. Classification and Structure

Phycobiliproteins are the major fluorescent proteins present in Spirulina that are responsible for light uptake. They are subdivided into phycoerythrin (PE), phycocyanin (PC), and allophycocyanin (APC), according to their pigment colors [41]. Phycocyanin is localized in phycobilisomes, a supramolecular protein complex that acts as an antenna of the photosynthetic apparatus at thylakoid level, attached to the thylakoid membrane [42] (Figure 1).

Phycocyanin, a high value protein, is constituted by one polypeptide alpha (α) ranging from 10 to 19 kDa and one polypeptide beta (β) ranging from 14 to 21 kDa that form a monomer. The subunit α is covalently linked with one phycocyanobilin and the subunit β has two phycocyanobilin attached [13]. Phycocyanobilin is a bilin chromophore of phytochromes with an open chain tetrapyrrole group that binds proteins through a thioether bond, and that confers the blue color to phycocyanin [13] (Figure 2). The monomers gather to form a ring-shaped trimer forming hexametric structures that confer stability to phycocyanin [13].

The efficient extraction of phycocyanin from Spirulina requires the disruption of the thylakoid membrane, which can be achieved through the various physical and chemical methods described below. The purity of phycocyanin is determined by A_620_/A_280_ ratio, being A_620_ for the absorbance of phycocyanin at 620 nm and A_280_ for the absorbance of other proteins at 280 nm. Moreover, the phycocyanin purity index will determine its market value and, consequently, its applicability. Indeed, phycocyanin is considered food, cosmetic, reagent, or analytic grade when A_620_/A_280_ is, respectively, greater than 0.7, 1.5, 3.9, or 4.0 [44]. The commercial value of phycocyanin can vary from 25 €/mg when partially purified to 200 €/mg for phycocyanin purity of 3.5 [45]. So, by improving the extraction, purification, and development of more stable formulations of phycocyanin, it can reach a high commercial value. The phycocyanin concentration (A) and yield (B) are obtained as described for Bennet and Bogorad [46], in which the absorbance at 652 nm corresponds to allophycocyanin.
(A) Phycocyanin (mg/mL) = (A615 − (0.474 × A652))/5.340
(B) Yield (mg/g) = (Phycocyanin × Solvent volume)/(Dried biomass)

### 3.2. Biological Functions and Applicability

The demand for replacing chemical and synthetic molecules by natural ones has been increasing over the years. There are several articles, based on in vivo and in vitro assays, reporting the biological functions of phycocyanin and, consequently, its applicability. Although phycocyanin from Spirulina has multiple functions and applications, Figure 3 and Table 2 summarize its main biofunctions described in the literature. As phycocyanin is a high value protein, it has been used in food as a dietary supplement or natural dye. Its utilization as food colorant in the US is also regulated by FDA [47], while in Europe, its use must comply with Regulation (EC) No 1333/2008 on food additives [48]. Phycocyanin extracts from Spirulina were classified as a GRAS ingredient in conventional food products (GRN 424) at levels up to a maximum of 250 mg/serving in foods and beverages [49]. Phycocyanin is also used in the cosmetic and biotechnology industries [27,50,51,52], and as a biochemical tracer, such as a fluorescent probe for cell analysis, in histochemistry, fluorescence microscopy, flow cytometry, and immunoassays [53].

The overproduction of reactive oxygen species (ROS) and the resulting oxidative stress occurs during the aging process and is a pathological feature of several diseases, such as neurodegeneration, inflammation, cancer, atherosclerosis, diabetes mellitus, and hyperpigmentation [107]. 

Romay et al. (1998) first evaluated the in vitro capacity of phycocyanin to scavenge hydroxyl radicals (^•^OH), alkoxyl radicals (RO^•^) by luminol-enhanced chemiluminescence, and zymosan activated human polymorphonuclear leukocytes, and deoxyribose assay, and by analysis of lipid peroxidation inhibition induced by Fe^2+^—ascorbic acid [54]. They also demonstrated that phycocyanin reduced the edema produced by an in vivo glucose oxidase-induced inflammation in mouse paw [54]. Other studies demonstrated the capacity of phycocyanin to scavenge superoxide anions (O_2_^•−^), singlet oxygen (^1^O_2_), hypochlorous acid (HOCl), peroxyl radical (ROO^•^), peroxynitrite (ONOO^−^), nitric oxide (^•^NO), and hydrogen peroxide (H_2_O_2_) [95,108,109]. Apoprotein has been proposed regarding the mechanisms that explain the way that phycocyanin stabilizes ROS [110]. Romay et al. (2000) [108] suggested that apoprotein scavenges HOCl by reacting with cysteine and methionine residues, but other authors demonstrated that tryptophan, tyrosine, and histidine are also able to scavenge ROO^•^ [111]. In vitro assays using DPPH, FRAP, and chelating activities demonstrated that treatment of phycocyanin inhibit lipid peroxidation and provide good metal chelation [55,56]. It was reported that 80 μg/mL phycocyanin decreased the UVB-induced production of ROS by 55% and reduced levels of matrix metalloproteinase (MMP)-1 and MMP-2 [59]. In in vivo studies, an increase in antioxidant activity was observed, too, after the administration of phycocyanin by catalase activity [58] and 1,1-Diphenyl-2-picrylhydrazyl (DPPH) free radical assay [57]. Superoxide dismutase (SOD) is an antioxidant enzyme that converts O_2_^•−^ into H_2_O_2_ and oxygen [58]. The effect of phycocyanin to increase SOD production was demonstrated in zebrafish [112], radiation-induced oxidative damage in C57BL/6 mice [113], and Balb/c mice models [58]. The mechanisms underlying this effect are poorly understood, but studies suggest that this may be explained by the ability of phycocyanin to scavenge free radicals and, thus, oxidative stress in cells [58]. Xu et al. [112] demonstrated that treatment of healthy zebrafish embryos with 5–320 μg/mL phycocyanin peptides had a protective effect on H_2_O_2_-induced damage by attenuating the production of ROS. The authors hypothesized that phycocyanin stabilizes and activates nuclear factor E2 related factor 2 (Nrf2) gene expression, leading to an increase in SOD and, consequently, a decrease in the production of ROS [112]. In another study, treatment of C57BL/6 mice with 200 mg/kg phycocyanin was shown to protect against radiation-induced oxidative stress by activating Nfr2 signaling, leading to an increase in SOD [113]. Administration of 500–100 mg/kg phycocyanin in Balb/c mice resulted in an increase in serum SOD activity (from 18 to 25–30 U/L). This study also suggested that phycocyanin has an effect against the effects caused by radicals, observed by an increase oin SOD activity [58]. Overall, the ability of phycocyanin to increase the activity of SOD by activating Nfr2 signaling likely contributes to its antioxidant effect and may be one of the mechanisms for its health benefits. The effect of phycocyanin in platelet aggregation was also evaluated. A study using human platelet suspensions demonstrated that 0.5–10 nM phycocyanin inhibited platelet aggregation through the attenuation of hydroxyl radicals in collagen-activated platelets [60]. On the other hand, treatment of mononuclear cells with 5 μM phycocyanin and 2 Gy radiation increased the activity of antioxidant enzymes, such as SOD, catalase, and glutathione-S-transferase [61]. Administration of 10 μM phycocyanin also demonstrated a protective ability against ischemia-reperfusion in a rat model of heart injury. The authors suggested that phycocyanin improved heart function recovery by suppressing the generation of free radicals by activation of ERK1/2 and suppression of p38 MAPK, Bax, and apoptotic markers [62]. In vitro treatment of human hepatocyte cell line L02 with 31–250 μg/mL protected against carbon tetrachloride-induced hepatocyte damage by attenuating the production of ROS and maintaining SOD activity [63]. Other studies demonstrated that in vitro treatment of 5–50 mM phycocyanin protected against oxalate-induced production of ROS in MDCK cells. Pre-treatment of phycocyanin also protected against mitochondrial membrane permeability and decreased the expression of phosphorylated JNK/SAPK and ERK1/2 [64].

Cancer is a pathological condition characterized by uncontrolled growth and division of cells that can invade normal tissues and spread throughout the body [114]. The role of phycocyanin in cancer has been demonstrated by its ability to block tumor cell proliferation, inhibit the cell cycle, and induce apoptosis and autophagy of these cells [115]. Phycocyanin was particularly associated with cell cycle arrest in the G0/G1 phase in breast cancer [65]. In vitro treatment of MDA-MB-231 cells with 55 μM phycocyanin reduced proliferation and caused G1 cycle arrest by downregulation of Cyclin E and CDK-2 and up upregulation of p21 levels [65]. The authors also observed that phycocyanin inhibited COX-2 and prostaglandin E production, suggesting that its effect could be explained by a decrease in the MAPK signaling pathway [65]. Phycocyanin was also involved in the activation of apoptotic pathways through the activation of caspase-9 and-3, which are responsible for DNA fragmentation and cell shrinkage, induction of poly [ADP-ribose] polymerase 1 (PARP-1) cleavage, and alteration of the Bcl2/Bax ratio. [67]. The role of phycocyanin in the alteration of the mitochondrial membrane potential in hepatocellular carcinoma cell lines was also demonstrated [66,67]. Treatment of these cells with 10–100 μM phycocyanin promoted cytochrome c release and caspase-3 activation, a decrease in membrane potential, and PARP cleavage [67]. The effect of phycocyanin in metastasis was demonstrated through downregulation of vascular endothelial growth factor A (VEGF-A), MMP-2, and MMP-9, which are required for tumor invasion and metastasis [68,69]. MDA-MB-232 and HepG2 cell line treatment with 5–40 μg/mL phycocyanin reduced the activity of MMP-2 and MMP-9 and reduced the mRNA expression of TIMP-2, but the mechanisms related to these results remain nuclear [68]. Treatment of 1,2-Dimethylhydrazine dihydrochloride-induced rat colon cancer animals with 200 mg/kg phycocyanin promoted a decrease in MMP-2, MMP-9, and MCP-1, which are all angiogenic factors upregulated in the model described [69]. The authors observed that phycocyanin interacts with active sites of VEGF-A receptors, leading to a decrease oin gene and protein expression of VEGF-A [69]. Other studies demonstrated that treatment of 5–40 μg phycocyanin enriched with selenium for 72 h disrupt the mitochondrial membrane potential, leading to a decrease in proliferation and an increase in apoptosis in melanoma and breast adenocarcinoma cell lines [70]. The beneficial effect of phycocyanin in drug resistance was also demonstrated [71]. Co-treatment of 1–100 µM phycocyanin for 12–48 h prevented the induction of multidrug resistance protein (MDR-1) by scavenging ROS and inhibiting COX-2 expression in the HepG2 cell line [71]. Interestingly, a synergistic effect of phycocyanin was observed in combination with chemotherapeutic agents and radiation. These results are summarized in Jiang et al. (2017) [115]. Phycocyanin has a high binding ability to cancer cells, so its use for co-localization of tumors in vivo has been suggested [116,117].

Activation of the immune system is a response mechanism of the organism to injuries and pathogens. However, the overexpression of immune mediators or ROS and excessive inflammation can lead to healthy tissue cell damage and inflammatory diseases [118]. Several studies have demonstrated the anti-inflammatory and analgesic effects of phycocyanin [72,73]. Phycocyanin suppressed inducible nitric oxide synthase (iNOS) and cyclooxygenase-2 (COX-2) induction of nitric oxide (NO) and prostaglandin E2, resulting in a decrease in tumor necrosis factor-α (TNF-α) production of and neutrophil infiltration in a rat model of induced inflammation [72,73]. Another study showed that phycocyanin inhibited nuclear factor-kappa B (NF-kB) activation and decreased the production of NO, iNOS, and TNF-α in a macrophage cell line stimulated with lipopolysaccharide (LPS) [73]. Additionally, in vivo experiments in mice have yielded similar results, showing that phycocyanin suppresses the synthesis of TNF-α and interferon-gamma (IFN-γ), which are pro-inflammatory cytokines, in a concentration-dependent manner [58]. One of the consequences of exacerbated inflammation is fibrosis, which is characterized by abnormal deposition and production of extracellular matrix and collagen, transdifferentiation of myofibroblasts, and epithelial-mesenchymal transition. Progressive fibrosis leads to pathological scarring and architectural alteration, resulting in damage and impairment of normal organ function [84]. Indeed, the phycocyanin effect in the prevention of fibrotic diseases has been demonstrated. Studies showed that phycocyanin downregulated the expression of vimentin, fibronectin, and collagen, upregulated the expression of E-cadherin, and suppressed epithelial-mesenchymal transition in cancer cell lines [74,76]. The role of phycocyanin in other human inflammatory diseases, such as acute myocardial infarction [99], atherosclerosis [98,100], liver [63,113] and lung injury [74,75], has also been described. In animal models of cardiovascular disease, phycocyanin reduced lipid, pro-inflammatory mediators, and acute myocardial infarction-associated enzymes [98]. Phycocyanin protected hepatocytes from free radical damage and inflammatory infiltration [63,99] and attenuated oxidative stress and DNA damage in a mouse model of radiation-induced liver damage [113]. In lung injury models, phycocyanin attenuated LPS-induced myeloperoxidase (MPO) activity, ROS formation, COX-2, and NF-kB activation, downregulated pro-apoptotic proteins, and downregulated anti-apoptotic proteins [74]. In another study on acute lung injury, phycocyanin was shown to decrease total protein concentration in bronchoalveolar lavage fluid (BALF), oxidative stress index, and proinflammatory mediator content, and to improve epithelial cell viability [75].

A balance between commensal microorganisms is essential to human health and well-being. The skin acts as a physical barrier that protects the human body from external damage. Consequently, an imbalance of skin microbiome, dysbiosis, can be related to systemic diseases [119]. The impact of antimicrobial resistance in global public health is worsening, leading to higher medical costs, prolonged hospital stays, and increased mortality. In fact, it is estimated that antibiotic resistance will be the first leading cause of death by 2050, resulting in 10 million deaths and a cost of $100 trillion worldwide [120]. Therefore, the development of novel natural molecules to replace antimicrobial synthetized compounds is urgently needed. Several studies have investigated the effect of phycocyanin on microbial growth, including clinically relevant bacteria. Mohite et al. (2015) [78] showed that 35 μg/mL phycocyanin decreased the growth of *Propionibacterium acnes* and *Staphylococcus epidermidis* (resident microbiome), and *S. aureus* (transient microbiome) can lead to acne [121]. It has been shown that 10% are *Escherichia coli* (66%), *Bacillus* sp. (59–60%), *Staphylococcus aureus* (85%), and *Salmonella* Typhi (20%). Another study with 25 μg/mL phycocyanin showed an attenuation of the inhibition zone of *Listeria monocytogenes* (22.1 mm), *S. aureus* (16.5 mm), *Yersinia ruckeri* (14.5 mm), *E. coli* (12.2 mm), and *Streptococcus iniae* (10.8 mm) [55]. Higher concentrations of phycocyanin, 320 μg/mL, impaired the growth of *S. aureus*, *Aeromonas hydrofila*, and *Salmonella Enteritidis* with zones of inhibition between 29.0 mm and 31.0 mm [79]. When 1000 μg/mL phycocyanin was used, the antibacterial activity was observed against *Pseudomonas aeruginosa* (18.0 mm), *Klebsiella pneumoniae* (16.0 mm), *E. coli* (13.3 mm), and *S. aureus* (9.3 mm) [80]. However, phycocyanin had no effect on the growth of *Acinetobacter baumannii*, *Enterococcus durans* [77], and *Enterococcus faecalis* [79]. Imbalanced colonization of *S. aureus* is present in 90% of patients with atopic dermatitis [81]. In addition, dysregulation between a phycocyanin extract can attenuate the symptoms of acne by reducing the growth of *P. acnes* (26.1 mm) and *S. epidermidis* (24.6 mm) [81], revealing its potential in the treatment of atopic dermatitis and acne. Moreover, it was observed that purified 40–80 μg/mL phycocyanin also inhibited the growth of relevant fungi, such as *Candida albicans*, *Aspergillus niger*, *Aspergillus flavus*, *Penicillium* sp., and *Rhizopus* sp. [82].

Neurodegeneration is a progressive and irreversible condition characterized by loss of structure and function of neurons in the central and peripheral nervous systems [122]. The properties of phycocyanin suggest that it may attenuate the progression of several neurodegenerative diseases, such as multiple sclerosis, Alzheimer’s disease, and Parkinson’s disease [123]. The role of phycocyanin has been demonstrated in the Experimental Autoimmune Encephalomyelitis (EAE)-induced multiple sclerosis. In this model, phycocyanin acted as a scavenger of peroxynitrite species, inhibited oxidative DNA damage, and inhibited lipid peroxidation, all of which are markers of multiple sclerosis [109,124]. Other studies have shown that phycocyanin promotes remyelination of damaged brain tissue [83,125]. Phycocyanin also attenuated the production of enzymes involved in Alzheimer’s disease, such as Aβ production [84]. Later studies showed that phycocyanin binds to the Aβ40/42 peptide, disrupting the key mechanism of Alzheimer’s disease [87]. Other reports showed that daily intraperitoneal injection of phycocyanin attenuated the production of pro-apoptotic (Bax, Bcl-2, caspase-9) and pro-inflammatory mediators (TNF-α, NF-κB, interleukin (IL)-6, IL-1β) [85,86]. Daily oral administration of phycocyanin for 4 weeks demonstrated an increase in the *Ruminococcaceae*, *Rikenellaceae*, and *Lactobacillaceae* families, which are intestinal bacterial strains associated with improvements in models of Alzheimer’s disease [126,127]. Studies in models of Parkinson’s disease have shown that phycocyanin protects against α-synuclein toxicity and amyloid-β fibril formation [84], oxidative stress response, and glutathione metabolism, which are pathomechanisms that occur in Parkinson’s disease [87]. In summary, based on its properties and evidence, phycocyanin has been proposed as a nutraceutical and complementary therapy for neurodegenerative diseases [122].

Diabetes mellitus is a chronic metabolic disease characterized by persistent hyperglycemia caused by insufficient insulin production. This induces glycation and leads to the formation of macromolecules that accumulate in the heart, blood vessels, eyes, and kidneys [27,89]. Some studies have demonstrated that phycocyanin is an antidiabetic agent and has an antiglycation activity that reduces the side effects of diabetes [89,90,91,92,128]. Phycocyanin promoted glucose uptake in an insulin-resistant cell line [77] and inhibited α-amylase and β-glucosidase in vitro, attenuating starch absorption [88,129,130]. Treatment with 100–200 mg/kg phycocyanin for 21 days [90] and 45 days [89] decreased blood glucose, total low-density lipoprotein cholesterol, and triglycerides in serum and the liver, increased high-density lipoprotein cholesterol, suppressed glycation markers, and maintained redox state by attenuating lipid peroxidation in diabetes-induced rodent models [89,90]. Oral administration of 50 mg/kg phycocyanin for 30 days decreased lipid levels, insulin resistance, and blood glucose concentration in a rat model of type II diabetes mellitus [92]. Thus, phycocyanin is suggested as a natural compound that increases insulin sensitivity and ameliorates insulin resistance [90,92].

Hepatoprotective effect was observed when phycocyanin significantly attenuated the cyclosporine-induced nephrotoxicity in rats by modulating the oxidative stress-mediated apoptotic pathway. [93]. In in vitro and in vivo assays, phycocyanin was revealed to have a protective effect against carbon tetrachloride (CCl_4_)-induced hepatocyte damage by reducing lipid peroxidation accumulation in liver microsomes [94]. Phycocyanin may also be able to inhibit inflammatory infiltration through its anti-inflammatory activities in CCl_4_-induced hepatic damage by inhibiting transforming growth factor (TGF-1) and hepatocyte growth factor (HGF) expression [63].

The administration of phycocyanin was revealed to prevent the progression and complications of chronic kidney disease. In vivo experiments revealed that phycocyanin (18 mg/kg) reduced cisplatin-induced oxidative stress levels in CD-1 mice while maintaining antioxidant enzyme activity to prevent renal damage [95]. In addition, it was reported that phycocyanin oral administration (300 mg/kg) protected Type 2 diabetes mice against oxidative stress and renal dysfunction. Phycocyanin protected against albuminuria and renal mesangial expansion in db/db mice and normalized not only the tumor growth factor-β and fibronectin expression but also the urinary and renal oxidative stress markers as well as the expression of NAD(P)H oxidase components [96]. In a study using human kidney 2 (HK-2) cells and male C57BL6 mice, it was also suggested that the mechanism involved in phycocyanin activity against cisplatin-induced nephrotoxicity may be, at least in part, the suppression of p-ERK, p-JNK, p-p38, Bax, caspase-9, and caspase-3 [97]. 

Studies have also reported the phycocyanin activity in the attenuation of cardiovascular disease. The development of cardiovascular disease is determined by the increased amount of ROS in the body. According to studies, after phycocyanin administration (50 mg/kg) in rats with acute myocardial infarction, the levels of ROS, Bax expression, and caspase-9 release were significantly reduced [99]. It was also suggested that the activation of HMOX1 and the heme catabolic pathway may represent an important mechanism for the reduction of atherosclerotic disease [98].

Obesity is essentially caused by the imbalance between intake and consumption. Few studies reported the anti-obesity activity of phycocyanin [101,102]. To be specific, phycocyanin was demonstrated to enhance the expression of endothelial nitric oxide synthase (eNOS) in the aorta under the stimulation of adiponectin, improving blood pressure levels and obesity [101]. It was also reported that phycocyanin could attenuate obesity, possibly by reducing adipogenesis in 3T3-L1 cells and HFD-induced obese mice [102]. Of note, recently it was demonstrated that phycocyanin can also improve fertility by partially increasing ovarian and oocyte quality in obese female mice by decreasing the number of early apoptotic cells and reducing the expression of H3K9me3 in oocytes. It was suggested that phycocyanin may provide a new and promising clinical treatment of obesity-related infertility in females [131].

Wound healing is a dynamic and complex process involving biochemical and physiological activities that lead to regeneration and replacement of injured tissue [132]. Wound healing begins immediately after injury with an increase in pro-inflammatory mediators and cells. This is followed by an increase in proliferation, transdifferentiation, and migration of fibroblasts to the injured site, production of dermal and epidermal cells, and synthesis of collagen and the extracellular matrix to re-establish the skin barrier [133]. The role of Spirulina extracts in fibroblast proliferation and wound healing has been described for years [128,132,133]. Recently, this function has been attributed to phycocyanin. Madhyastha et al. (2008) [103] first described the dermal wound healing effect of phycocyanin in mice by promoting fibroblast proliferation and migration via activation of cyclin-dependent kinase (cd)K1, cdK2, urokinase-type plasminogen activator (uPA), and PI3K signaling pathways. They also demonstrated that 50–75.0 μg/mL phycocyanin increased the expression of vital mediators in wound healing, promoted proliferation, and increased migration and wound healing, leading to tissue regeneration in in vivo models [103,104].

Melanin, the major component of skin color, is produced in melanocytes by melanogenesis and protects the skin from environmental hazards such as UV radiation and pollution [105]. The melanogenesis process occurs in melanosomes, which contain key tyrosinase enzymes that catalyze and regulate the production of pigments such as melanin [134]. Extreme UV exposure and ROS formation can stimulate the overproduction of tyrosinases through activation of microphthalmia-associated transcription factor (MIFT) [134,135]. Excessive production and the deposition of melanin leads to skin pigmentation, post-inflammatory hyperpigmentation, and other skin and neurodegenerative disorders [135], such as freckles, age spots, melasma [136], Parkinson’s disease, and Alzheimer’s disease [137]. It was reported that 0.2 mg/mL phycocyanin increased intracellular cAMP levels to phosphorylate ERK1/2, resulting in the degradation of key transcription factors of melanogenesis (such as MITF) and, consequently, a decrease in tyrosinase activity [106]. Phycocyanin also regulated the tyrosinase gene expression by downregulating p-38 MAPK-regulated CREB activation [106]. In another study, a peptide library obtained from in silico enzymatic digestion of phycocyanin from *A. platensis* was shown to have tyrosinase inhibitory potential [105].

Despite evidence of phycocyanin’s properties in different diseases in in vivo and in vitro studies (Table 1), ongoing clinical trials are based only on Spirulina and its extracts. According to Liu et al. (2022) [127], four clinical trials are currently underway, three of which have been published in articles and the other only on the internet. Mahendra et al. (2013) [138] reported the beneficial effects of a Spirulina gel in the treatment of periodontitis. The authors believe that the anti-inflammatory property of the gel is due to phycocyanin [138]. In 2015, Patil et al. (2015) [139] demonstrated the effect of Spirulina in oral submucosal fibrosis by improving the degree of mouth opening, oral ulcers, and erosions. Ge et al. (2019) [140] described that Spirulina attenuated myelosuppression and improved immunity in patients undergoing chemotherapy. One clinical study aimed to assess the antioxidant efficacy of Spirulina on LDL and lipid metabolism on patients with metabolic syndrome by using phycocyanin concentrated fresh Spirulina water extract. The study was concluded but no results were reported (NCT02817620). Currently, another study is recruiting patients with metastatic gastric cancer to evaluate the effect of phycocyanin against oxidative stress in chemotherapy-induced peripheral neuropathy (NCT05025826). Due to the lack of official human clinical trials using phycocyanin, there is an urgent need to share knowledge of Spirulina with clinicians to improve the efficacy of treating human diseases.

Furthermore, several methods of extraction and purification of phycocyanin recovered from Spirulina have been reported. The selection of the best method and respective conditions will depend on several aspects, considering the time, cost, and yield of the processes. The intended amount of phycocyanin, small- or large-scale production, and the final application of the product are also determining factors in the selection of the most suitable methods to be used. 

### 3.3. Phycocyanin Extraction Methods

Several parameters must be considered for an efficient phycocyanin extraction, such as cell disruption, form of biomass (fresh or dried), temperature, light intensity, pH, solvent type, biomass/solvent ratio, and the use of preservatives [4]. The membrane of Spirulina consists of four layers, the outermost of which is covered by protein fibrils, a peptidoglycan layer that gives rigidity to the cell, and an innermost fibril layer [42]. The methods used to disrupt the cell membrane are critical factors for high extraction yield and phycocyanin purity (Figure 4 and Table 3). The best method is the one that selectively releases the target with the least energy input [4]. Several methods have been described in the literature, but only the freeze/thaw, mixing/homogenization, bead milling, ultrasonic, and electric field, high-pressure homogenization, and enzymatic extraction methods have been described with reliable purity values and a comparison of the best conditions and parameters (Table 3).

#### 3.3.1. Cell Disruption Methods

##### Freeze/Thaw Cycles

Freeze/thaw cycles are a method that promotes continuous damage to the plasmatic membrane [156]. The number of cycles and the optimal extraction ratio to improve the purity and the yield of phycocyanin are not precisely described in the literature. Some authors described that five and six cycles lead to the release of contaminating proteins but that four cycles is the best condition [141]. Other studies demonstrated that three cycles are the best strategy to improve phycocyanin extraction [142]. On the other hand, a wide range of solvent ratios (from 1:200 to 1:6) have been used without appropriate optimization [141,143,144,152,157]. The choice of solvent is an important parameter to consider in this technique. Of the solvents evaluated, sodium phosphate buffer (pH 7.0) [141,142,152] proved to be the most suitable, leading to a higher extraction yield (217.18 mg/g) [143] and purity (0.87) [142]. However, repeated freezing and thawing cycles are time and energy consuming, making this method suitable only for the laboratory scale [141].

##### Mixing and Homogenization

Mixing and homogenization are simple extraction methods with low and high stirring intensity, respectively. Using 1.71% biomass/solvent ratio, a 6237.66 homogenization rate, and 15 min of extraction time, the yield of phycocyanin obtained was 67.61 mg/g [145]. Extraction yields of 103.07 mg/g and purities of 0.67 were obtained using oven-dried biomass preparation at 70 °C for 4 h and extraction at 25 °C for 24 h by 0.01 M phosphate buffer using an homogenization assisted method at 0.02 g/mL biomass concentration [146]. However, cell debris can be released into the medium [145], and the temperature rise during mixing and homogenization reduces the purity of the extract [142]. Therefore, this method is not suitable for the industrial scale.

##### Bead Milling

Bead milling is a mechanical method in which the cell membrane is broken using high-speed beads [158]. The optimal conditions for this method are challenging to predict. However, it has been shown that low-density beads (e.g., glass) are more suitable for low viscosity media [159] and that a high degree of bead filling (80–85%) allows higher disruption rates [160]. It is a simple method with a duration between 5 min [147] and 4 h [148], a high biomass disruption capacity (60–150 g/L), a low energy input [161], and a high extraction yield (217.14 mg/g) [149], making it well suited to industrial application. On the other hand, the purity achieved is low (0.46) [149], requiring purification steps to remove cell debris [159].

##### Ultrasonic

Ultrasound treatment exposes the biomass to ultrasonic waves that cause thinning of cell membranes and disruption of cells [162]. Extreme power intensities can lead to intense and rapid membrane disruption and high temperatures, resulting in poor purity. Therefore, control of these parameters is critical in ultrasonic extraction. Temperature can be controlled using water jacket extraction chambers coupled to a water bath. Low extraction times and moderate amplitude conditions (40–60%) can be achieved using ultrasonic horns [155]. Studies with ultrasonic extraction [146,151] resulted in phycocyanin yields ranging from 17.20 mg/g [151] to 90 mg/g [146] and purity ranging from 0.67 [146] to 0.93 [140]. İlter et al. (2018) [145] obtained a yield of 98.84 mg/g phycocyanin using 1% biomass/solvent ratio, 60% amplitude, and a 16.23 min of extraction time [145]. Liao et al. (2011) [151] described phosphate-buffered saline (PBS) soaking as a more efficient (yield of 17.2 mg/g and purity of 0.93), simpler, and less expensive technique compared with ultrasonic extraction (yield of 18.2 mg/g and purity of 0.65). However, ultrasound was proposed by Vernès et al. (2019) [163] as an efficient method for the extraction of phycocyanin from Spirulina proteins. They developed an environmentally friendly method that can be transferred from laboratory to pilot and industrial scales [163].

##### Electric Fields

Extraction by electric fields is based on the application of short electric pulses that promote the formation of pores in the cell membrane and increase its permeability by electroporation [164,165]. The electric pulse can be pulsed or moderate if the electric field intensity is at least 1000 Vcm^−1^ or at most 1000 Vcm^−1^ [164,165]. The intensity of the electric field, the pulse duration and shape, and the duration of the electric field affect phycocyanin extraction. Extraction of phycocyanin by a pulsed electric field allows selective extraction. This method leads to the release of intracellular compounds without destroying the cell structure. Consequently, some internal compounds are not released, requiring an optimized diffusion step from 15 min to 3 h [147,149,150] after the pulsed electric field. With a moderate electric field, the size of the pores formed is smaller. The optimal conditions for a higher yield in phycocyanin extraction were 40 °C, 25 kV/cm and 150 μs of treatment [149]. Nevertheless, the higher phycocyanin concentration (151.94 mg/g) and purity (0.51) were achieved with pulsed electric fields [149].

##### High-Pressure Homogenization

The high-pressure homogenization (HPH) is a physical method that uses high pressure to homogenize the sample and break down the cell wall of the Spirulina, allowing the release of the phycocyanin pigment. Recently, HPH has been implemented for the phycocyanin extraction with the reported yields being significantly higher than those of conventional extractions [153]. Studies using HPH at pressures between 1400 and 2000 bars under phosphate solvent resulted in phycocyanin yields between 113.50 mg/g and 291.90 mg/g [152,153]. Pressure is a crucial factor in HPH expression but the optimization of HPH extraction also involves the amount of algae (biomass/solvent ratio), temperature, and number of passes through the HPH equipment [154]. Compared to the use of electric fields or sonication, HPH is a simpler and scalable process that can be easily integrated into industrial production processes.

##### Enzyme-Assisted Methods

The enzymatic extraction is another method that is gaining attention due to its potential to be a more efficient and eco-friendly alternative to traditional extraction methods. The extraction using enzymes can be performed at low temperatures and pressures, which can help to preserve the stability and quality of phycocyanin. In different studies, the lysozyme was used to digest the polysaccharides present in the cell wall matrix of dry Spirulina. Recovery values between 20 mg/g to 25 mg/g and a purity between 0.8 and 0.9 were achieved [166]. Incubation of phycocyanin with 0.6% lysozyme for 16 h resulted in a purity of 1.19 with a yield of 82.07 mg/g and recoveries of 68.96%. Ultrasound with enzyme-assisted extraction resulted in the highest phycocyanin yield of 92.73 mg/g dry biomass, with 77.92% extraction efficiency and a purity of 1.09 [167]. In another study, an effective extraction of phycocyanin was achieved using Collupulin protease in combination with the application of a pulsed electric field [167]. The combination of a pulsed electric field with enzymes can be a promising technology for phycocyanin extraction.

### 3.4. Key Parameters for Phycocyanin Extraction and Stability

#### 3.4.1. Biomass Form

Dry biomass is often used for phycocyanin extraction, although some studies have described the use of fresh biomass [149,150,168]. There is no evidence of the technique achieving higher extraction yields, but one study demonstrated that rehydration of the powder for 2 h in the extraction buffer results in higher yields [155]. Conventional drying at temperatures up to 60 °C is not recommended as it can lead to degradation of phycocyanin [145].

#### 3.4.2. Temperature

Higher temperatures are important to extract the intracellular compounds into the medium, but phycocyanin loses its stability at high temperatures. In addition, extreme light exposure and low pH could contribute to degradation of phycocyanin [4]. Temperatures between 25 °C and 47 °C and a pH of 6 were described as suitable conditions [169]. Wu et al. (2016) [57] demonstrated that phycocyanin was not degraded up to 45 °C, but degradation increased between 47 °C and 69 °C [57,169,170,171] and increased further after 70 °C [56,169,172]. Therefore, a temperature up to 45 °C is the optimal temperature to maintain the stability of phycocyanin and prevent its degradation [173].

#### 3.4.3. Light

Light quality and intensity affect algae growth, photosynthesis, and, consequently, phycocyanin production [174]. Each species of algae grows under specific conditions and types of light. Regarding the effect of light spectra in Spirulina, Walter et al. (2011) [175] verified that photobioreactors manipulated with red light achieved higher photosynthetic efficiency and phycocyanin compared to blue light [175]. Wicaksono et al. (2019) [157] also demonstrated that the red spectrum produced the greatest amount of phycocyanin [157]. The correlation between light and temperature [176] also impacts phycocyanin degradation. A study using light intensities of 50 and 100 µmolm^−2^s^−1^ at 25 °C demonstrated that phycocyanin concentration decreased with time in a dose-dependent manner [177]. Moreover, protein concentration decreased by approximately 20% after 36 h of continuous exposure to 100 µmolm^−2^s^−1^, and degradation was higher at this intensity than at 50 µmolm^−2^s^−1^ [177]. Therefore, the best storage conditions are found in the dark [56].

#### 3.4.4. pH

The pH of the medium is important to maintain the stability of phycocyanin and to preserve its spectral properties and characteristic blue color. Using ultrafine fibers of polylactic acid or polyethylene oxide as pH indicators [9], it has been shown that precipitation of contaminating proteins and degradation of phycocyanin are higher at pH 3.0–4.0 due to changes in their conformation [56,168]. The best pH conditions were found at values between 5.5 and 7.0 [56,169,170,171,173,178].

#### 3.4.5. Type of Solvent

The solvent also affects the solubility of phycocyanin due to its ionic strength, which affects the protein structure [4]. The choice of solvent depends on the range of pH stability of phycocyanin. Some authors have shown that sodium phosphate buffer (pH 7.0) leads to a higher extraction yield compared to distilled water [142,146,151,152]. Others showed that a higher yield of phycocyanin is obtained when distilled water [179,180] or potassium phosphate buffer (pH 6.5) [181] or CaCl_2_ (pH 7.4) [145] were used. In contrast, a lower extraction yield was obtained with acetate buffer (pH 5.0) [142].

#### 3.4.6. Biomass/Solvent Ratio

Higher biomass/solvent ratios are associated with higher extraction yields but may result in lower phycocyanin content [4]. In a range of ratios between 1:6 and 1:10, studies using sodium-phosphate buffer (pH 6.8) demonstrated a higher extraction yield at a 1:10 ratio and a higher purity at a 1:6 ratio [141,155]. Namely, values of extraction yield were observed of 53.84 mg/g and a purity of 0.6 (extraction yield of 52.11 mg/g) at 1:6 ratio [130]. In a study in which different extraction methods and surfactants (Triton X-100, Tween 20 and Tween 80) were compared, it was also observed that higher values of purity at 1:6 ratio 0.67–1.09 (extraction yield ranging 52.92–92.73 mg/g of dry biomass and 50.35–77.92% of extraction efficiency) compared with 1:10 ratio (purity of 0.68–0.9 and extraction yield of 65.14–99.63 mg/g) [155]. This could be since more interfering proteins can be extracted with a higher volume of solvent [141]. In other work, extraction ratios of 1:50, 1:25, and 3:50 were investigated using sodium-phosphate buffer (pH 7.0) and the highest extraction yield and phycocyanin purity that were obtained were at 1:50 (purity-1.2; extraction yield—~80%) [146]. However, ratios of 1:25 (purity-0.46; extraction yield-3.68 mg/mL) [142], 1:20 (extraction yield-41.90 mg/g) [144], 1:15 (purity-0.8) [152], and 1:50 [178] were also used and reported. The ratio of biomass/solvent should be selected to minimize the costs associated with the process.

#### 3.4.7. Preservatives

Despite all the care taken in the extraction of phycocyanin, it is very sensitive to environmental stresses. Therefore, preservatives are used to increase the stability of phycocyanin and maintain its bioactivity [182]. 

Small saccharides such as sugars promote protein-solvent interactions and support protein folding [183]. The type of sugar and its concentration affect phycocyanin stability. The addition of glucose (20–40% *w*/*w*), fructose (10–15% *w*/*w*), and sucrose (70% *w*/*w*) increases the half-life of phycocyanin [169,183,184]. Chaiklahan et al. (2012) [169] used 20% *w*/*w* glucose and saccharose and observed that phycocyanin was more than 62% preserved after 15 min at 60 °C compared to 47% without preservative. Wu et al. (2016) [56] demonstrated that sodium chloride is a suitable agent to protect phycocyanin degradation, but that it becomes turbid at concentrations of 5% *w*/*w* [169].

Citric acid at a concentration of 0.4% *w*/*w* increases the concentration of phycocyanin to 19% instead of the 11% observed without the use of preservatives [185]. Another study showed that citric acid preserved 67% phycocyanin after 45 days compared to 3% without the use of preservatives [186].

Crosslinkers such as formaldehyde and glutaraldehyde create networks between polymer chains and proteins that prevent phycocyanin degradation, but they are toxic and cannot be used in the food industry [187].

Natural polymers have also been evaluated for phycocyanin protection. The addition of polysaccharides increases the viscosity of the medium, so it cannot be used for some applications. Studies show that whey protein, egg white, pea protein, and carrageenan provide chromophore protection, which protects phycocyanin from degradation and maintains its blue color [188,189].

Encapsulation of phycocyanin in nanofibers, microparticles, or nanoparticles is also commonly used to prevent phycocyanin degradation at unstable temperatures and pH [190,191,192,193,194,195].

Nevertheless, the development of new techniques and options to obtain and maintain phycocyanin with high purity is a future challenge. The use of microfluidics or supercritical fluid technology in phycocyanin formulation has been proposed as a promising alternative [196].

### 3.5. Phycocyanin Purification Methods

Optimization of phycocyanin purification is important to increase extraction yield and purity, improve market value, and reduce costs involved in the process [130]. Among the most used and described methods, as shown in Table 4, is a combination of precipitation with ammonium sulphate (AS), membrane filtration (MF), and chromatography (Figure 5).

The AS precipitation is a simple and cheap method to improve the recovery and purity of phycocyanin [44,202]. Different concentrations of AS have been evaluated, and the better purity (2.11) and recovery values (86%) of phycocyanin were obtained with 50–65% AS [197,203,204,205]. Moreover, it was reported to be more efficient for phycocyanin recovery than other salts, such as sodium dihydrogen phosphate, magnesium sulphate, and dipotassium hydrogen phosphate [206]. On the other hand, AS precipitation takes a long time, and the separation of phycocyanin from the salt does not occur properly [130]. For this reason, phycocyanin purification protocols have been optimized by adding a dialysis or microfiltration or ultrafiltration step [130]. Chromatographic methods include ion exchange chromatography (IEC), gel filtration chromatography, hydrophobic interaction membrane chromatography, and membrane chromatography. The most used method in the literature is IEC. With this method, it is possible to obtain phycocyanin in high purity and recovery [130].

Starting from a crude extract with a purity of 0.75–0.97, studies demonstrated an increase in purity after precipitation with 60–65% AS to 1.50 and 2.11, respectively [197,204]. After dialysis and IEC, the purity of phycocyanin increased to 4.05–5.59 and recoveries were 67% [197]. Other authors performed precipitation with 50% AS followed by dialysis, ultrafiltration with a molecular weight cut-off (MWCO) of 50 kDa, and IEC with pH gradient [44] or linear salt gradient elution [202]. They achieved purity values above 4.0, but the phycocyanin yield was higher in the study using IEC with a pH gradient (80% vs. 68%) [41,189]. Prabakaran et al. (2020) [88] performed purification with 65% AS followed by 65% dialysis with 12–14 kDa membranes, microfiltration, IEC, and HPLC with a reverse column. They expressed purity in percentage terms, reaching values of 92% and a protein content of 53% [88]. In some studies, chitosan or activated charcoal were used instead of AS for precipitation [151,198]. After this step, phycocyanin purity and yield of 1.52 and 96% were obtained in one study using chitosan and 2.78 and 85% using activated charcoal, respectively. However, after IEC, the authors achieved a purity of 4.3 but a yield of less than 50% [151]. Another study using 0.24% *w*/*v* chitosan and 8.4% *w*/*v* activated charcoal obtained a purity of 3.14 and a recovery of 79% [198]. Other authors used a polyvinylidene fluoride (PVDF) membrane, and after precipitation with AS, the purity increased from 0.9 to 3.7 [201]. After chromatography, they used two hydrophobic interaction membrane chromatography steps and obtained a phycocyanin purity of 4.2 and a yield of 67% [201]. Chen et al. (2019) [199] used a single method with 10% and 1% *dw*/*v* stirred fluidized bed IEC. Using 10% *dw*/*v* or 1% *dw*/*v*, they obtained a purification factor of 3.0 and 2.7, respectively, and a yield of 64.3% and 90.6%, respectively [186]. Amarante et al. (2020) [200] obtained a phycocyanin purity between 3.5 and 4.2 and recoveries between 32% and 49% using a single IEC step with pH gradient elution. Patil et al. (2006) [207] described a purification method for phycocyanin using ion-exchange chromatography, which resulted in a purity of 5.22. 

Most of the proposed methods for obtaining reagent grade or analytical grade phycocyanin require multiple steps and are costly. Therefore, there is an urgent need to optimize these methods to reduce the cost and time required for industrial-scale application.

## 4. Conclusions

Spirulina is becoming increasingly popular as a healthy dietary supplement. It is considered a safe and high-quality food and is used as a substitute for animal or plant foods due to its impressive nutritional profile, especially its high protein content. In addition, several of its biological properties are attributed to its high phycocyanin content. This pigment-protein complex is used in the food, cosmetic, and pharmaceutical industries due to its excellent biological activities.

This review focuses on the reported biological properties of phycocyanin, such as its antioxidant, anti-inflammatory, antimicrobial, anticancer, anti-neurodegenerative, antidiabetic, hepatoprotective, nephroprotection, cardiovascular protection, anti-obesity, melanin overproducing protection, and wound healing activities. Additionally, it describes the optimal conditions for maximizing phycocyanin production from Spirulina cultures, namely, a temperature of 30 °C, a pH between 10 and 10.5, a light intensity of 300 μmol photon/m^2^/s, and a medium rich in several macro- and micronutrients. The sensitivity of phycocyanin to environmental conditions influences its use for biological functions and the development of new formulations and products. Thus, this review also highlights the main factors affecting the chemical degradation of phycocyanin and the strategies available to improve its stability. In particular, the cell disruption method must allow the release of the total pigment content while maintaining stable medium conditions. According to major studies, phycocyanin is best preserved when the temperature is below 45 °C, the pH is between 5.5 and 6.0, storage is in the dark, and preservatives such as mono- and di-saccharides, citric acid, crosslinkers, and natural polymers are used. The recovery of phycocyanin with high purity is essential to increase its market value. The best purification conditions using ammonium sulfate, filtration methods, and chromatographic techniques are also presented.

Finally, further research is needed to develop cost-effective and label-friendly methods and novel preservation molecules to improve the stability and purity of phycocyanin in the food, cosmetic, and pharmaceutical industries on a large scale. In addition to phycocyanin, other secondary metabolites of Spirulina may also have the potential to be used in other industries such as the food, biofuels, biomaterials, biofertilizer, and animal feed industries, contributing to industrial symbiosis and a circular economy. Hence, the scale-up of these processes also needs to be improved.

## Figures and Tables

**Figure 1 pharmaceuticals-16-00592-f001:**
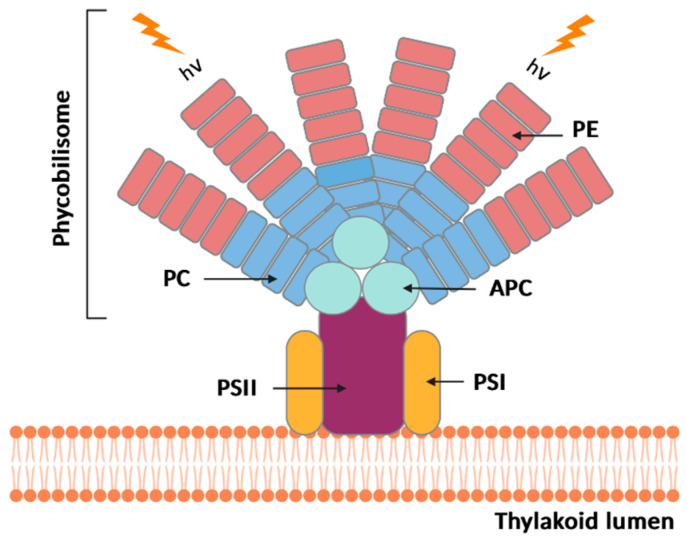
Structural organization of phycobilisomes. This complex is constituted by phycoerythrin (PE), phycocyanin (PC), and allophycocyanin (APC), organized to transfer energy (hν) unidirectionally to the reaction center in a highly efficient way. The cascade begins with PE to PC and APC and finally to the reaction center in photosystems (PS) II and I. Adapted from Pagels et al. (2019) [43].

**Figure 2 pharmaceuticals-16-00592-f002:**
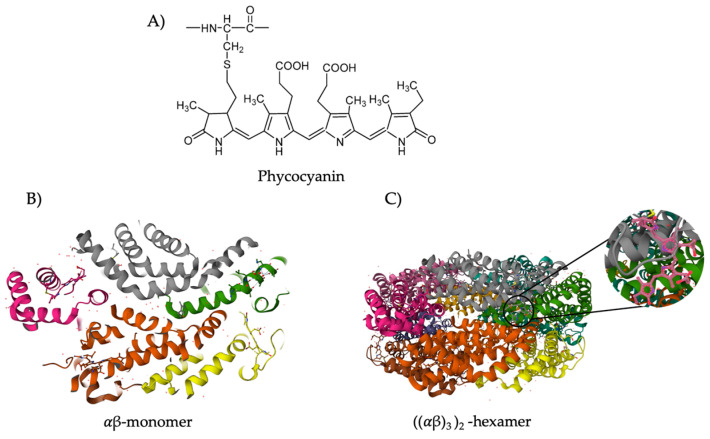
Chemical, (**A**), and protein, (**B**,**C**) structure of phycocyanin from *Spirulina platensis*. (**B**) Represents the αβ monomer, and **C**) represents the three-dimensional (α_3_β_3_)_2_ hexamer structure of phycocyanin. The α subunit is represented by green, orange, and pink and the β subunit by grey, yellow, and brown. (**C**) Highlights a phycocyanobilin chromophore. α- and β- subunits have one and two phycocyanobilin, respectively. α: alpha, β: beta. Adapted from Yuan et al. (2022) [13] and Wu et al. (2016) [42].

**Figure 3 pharmaceuticals-16-00592-f003:**
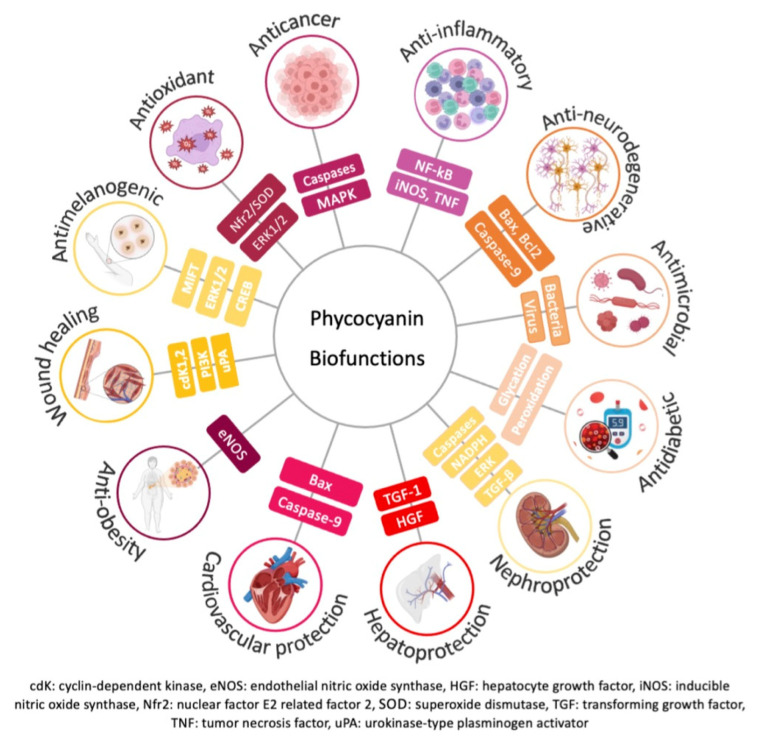
Biological functions of phycocyanin recovered from Spirulina.

**Figure 4 pharmaceuticals-16-00592-f004:**
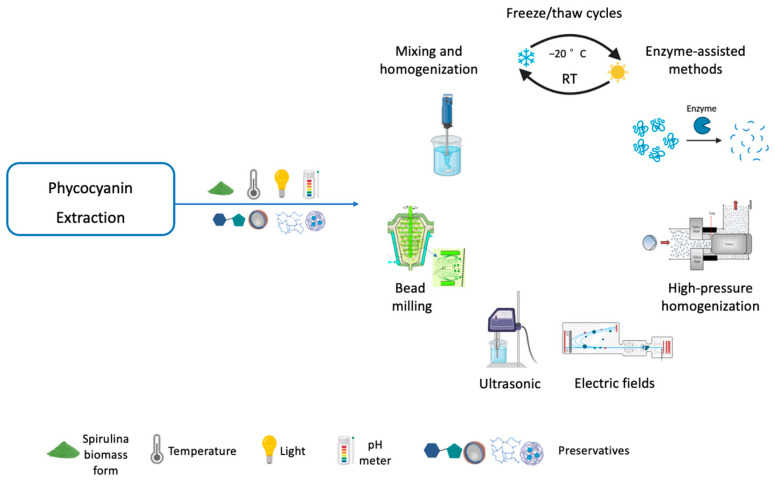
Extraction methods for phycocyanin recovered from Spirulina and use of preservatives to maintain its stability. Cell disruption methods include freeze/thaw cycles, mixing and homogenization, bead milling, ultrasonic, electric fields, high-pressure homogenization, and enzyme-assisted method. Biomass form, temperature, light, pH, type of solvent, and biomass/solvent ratio are key conditions during phycocyanin extraction. RT: room temperature.

**Figure 5 pharmaceuticals-16-00592-f005:**
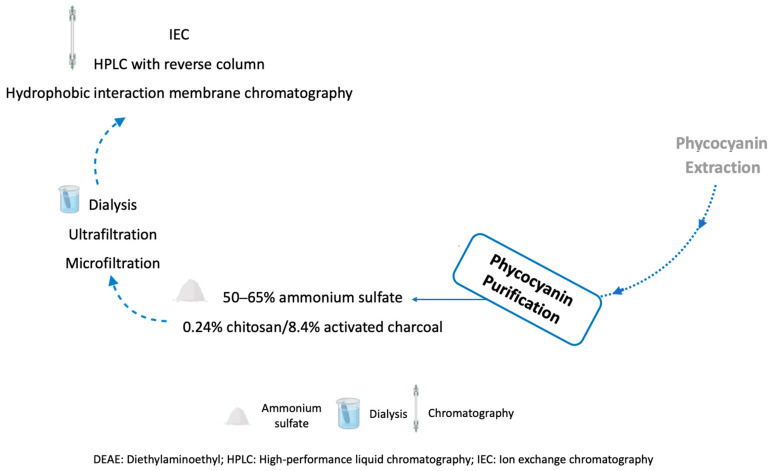
Purification methods for phycocyanin recovered from Spirulina. Purification involves a combination of techniques, and the main ones described are ammonium sulfate precipitation, membrane filtration, and chromatography.

**Table 1 pharmaceuticals-16-00592-t001:** Description of the composition of Spirulina, including its content in vitamins, minerals, and amino acids present in 10 g of Spirulina [22,28,29].

Compound	Amount	Ref.	Compound	Amount	Ref.
**General composition (in dry weight Spirulina)**			Inositol	6.4 mg	[28]
Proteins	Folic acid	[22,28]	Pantothenic acid	10 μg	
Carbohydrates	Biotin		Folic acid	1.0 μg	
Total lipids	5.0–6.0%		Biotin	0.5 μg	
Fibers	3.6–6.0%		**Minerals (*per* 10 g Spirulina)**		
Minerals	7.0–13%		Potassium	140 mg	[28]
**Vitamins (*per* 10 g Spirulina)**			Sodium	90 mg	
Vitamin A	23000 IU	[28]	Calcium	70 mg	
Vitamin B1, B2, B3	0.4–1.4 mg		Phosphorus	60 mg	
Vitamin B12, B6	20–60 μg		Magnesium	40 mg	
Vitamin C	0.8 mg		Iron	15 mg	
Vitamin D	1200 IU		Manganese	0.5 mg	
Vitamin E	1.0 mg		Zinc	0.3 mg	
Vitamin K1, K2	200 μg		Cooper	120 μg	
Germanium	60 μg	[28]	Histidine	100 mg	[29]
Iodine	55 μg		Tryptophan	90 mg	
Chrome	25 μg		**Nonessential amino acid (*per* 10 g Spirulina)**		
Selenium	10 μg		Glutamic acid	910 mg	[29]
**Essential amino acid (*per* 10 g Spirulina)**			Aspartic acid	610 mg	
Leucine	540 mg		Alanine	470 mg	
Valine	400 mg		Arginine	430 mg	
Isoleucine	350 mg		Glycine	320 mg	
Threonine	320 mg		Serine	320 mg	
Lysine	290 mg		Tyrosine	300 mg	
Phenylalanine	280 mg		Proline	270 mg	
Methionine	140 mg		Cystine	60 mg	

**Table 2 pharmaceuticals-16-00592-t002:** Biological roles of phycocyanin recovered from Spirulina and description of its main functions, dose, or concentration, and methods used to analyze this effect.

Role	Effect	Dose or Concentration (Route of Administration)	Analysis Methodology	Ref.
Anti-oxidative	Scavenging of free radicals, lipid peroxidation inhibitor and metal chelator	1–3 mg/mL	Luminol-enhanced chemiluminescence	[54]
		0–0.16 mM	Deoxyribose assay	
		8–20 mg/mL	Inhibition of liver microsomal lipid peroxidation induced by Fe-ascorbic acid	
		50–200 mg/kg (oral)	Glucose oxidase-induced inflammation in vivo	
		62.34 mg/g	DPPH, FRAP and Fe^2+^—chelating activity	[55]
		0.125–2.00 mg/mL 0.3125–5.00 mg/mL	ABTS and DPPH	[56]
		10–100 μg/mL	DPPH	[57]
	Serum antioxidant	200–1000 mg/kg (oral)	SOD and catalase activity in vivo	[58]
	Attenuation of MMPs and ROS	20–80 μg/mL	MMP-1 and MMP-9 and DCFDA staining	[59]
	Attenuation of platelet aggregation by decreasing hydroxyl radicals	0.5–10 nM	Electron Spin Resonance Spectrometry	[60]
	Increase of antioxidant enzymes	5 μM + 2 Gy radiation	RANDOX kit	[61]
	Attenuation of ROS	10 μM (rat heart perfusion)	Electron paramagnetic resonance spectroscopy	[62]
	Attenuation of ROS, MDA and GSH, and maintenance of SOD activity	100–400 mg/kg (intraperitoneal injection) 31–250 μg/mL	DCFDA staining and histopathologic analysis	[63]
	Free radical scavenger	5–50 mM pre-treatment for 1 h and then co-treatment	DCFDA staining	[64]
Anticancer	Cell cycle arrest in G0/G1, attenuation of proliferation and stimulation of apoptosis	1–20 μM	Propidium iodide, annexin V-PE, 7-AAD, proliferative, and apoptotic markers	[65]
		10–100 μM	MTT assay, cytochrome c, ethidium bromide	[66]
	Alteration of the mitochondrial membrane potential	10–100 μM	Rhodamine 123	[67]
	Attenuation of MMPs	5–40 μg/mL	MMP-1, MMP-2, MMP-9, TIMP-1, TIMP-2	[68]
	Attenuation of metastasis	200 mg/kg (oral)	MMPs, VEGF-A and HIF-1α, activity of MMPs and HIF-1α	[69]
	Stimulation of mitochondria-mediated apoptosis	5–40 μg	Depolarized mitochondria, apoptotic, and proliferative markers	[70]
	Drug resistance by preventing the induction of multidrug resistance protein	1–100 μM	ROS production and COX-2 expression	[71]
Anti-inflammatory	Attenuation of pro-inflammatory mediators and neutrophil infiltration	30–50 mg/kg	TNF-α, IL-1β, IL-10, nitrite, nitrate, PGE2, COX-2, iNOS, MPO, and NF-kB activity	[72]
		0–250 μg/mL		[73]
	Attenuation of lung injury	50 mg/kg 100–400 mg/kg (intraperitoneal injection)	Lung injury, nitrate/nitrite, pro-inflammatory cytokines in BALF, MPO and NF-kB activity, iNOS, COX-2, lung edema, proapoptotic proteins	[74,75]
	Prevention of fibrosis	10–50 μg/mL 0–200 μg/mL	Nrf2, NQO-1, EMT evaluated through the expression of vimentin, type-1-collagen, fibronectin, α-SMA, N-cadherin, and E-cadherin	[76,77]
Antimicrobial	Decrease the growth of *Escherichia coli*, *Bacillus* sp., *Staphylococcus aureus*, and *Salmonella* Typhi	35 μg/mL	Disc diffusion assay and determination of MIC. Comparation with Antibiotic Assay Medium (Himedia).	[78]
	Attenuate the growth of *Listeria monocytogenes*, *S. aureus*, *Yersinia ruckeri, E. coli*, and *Streptococcus iniae*	25 μg/mL	Agar well diffusion assay, MIC and MBC. Comparation with Tetracycline, Amikacin, and Doxycycline	[55]
	Impair the growth of *S. aureus*, *Aeromonas hydrofila*, and *Salmonella* Enteritidis. No effect in *Enterococcus faecalis*	320 μg/mL	Agar well diffusion method and turbidity liquid media assay. MIC and turbidity at 600 nm	[79]
	Antibacterial activity against *Pseudomonas aeruginosa* MTCC 1034, *Klebsiella pneumoniae* (ESBL-KP) ATCC 700603, *E. coli* (ATCC 25922), and *S. aureus* ATCC 25,923 (MRSA). No effect on *Acinetobacter baumanii*, *Enterococcus durans* (P502).	1000 μg/mL	Mueller–Hinton Agar plates and MIC using broth microdilution method	[80]
	Attenuation of acne symptoms and reduction of *Propionibacterium acnes* and *Staphylococcus epidermidis*	10% extract	Disc diffusion method and MIC	[81]
	Inhibition of the growth of *Candida albicans*, *Aspergillus niger*, *Aspergillus flavus*, *Penicillium* sp., and *Rhizopus* sp.	40–80 μg/mL	Agar block method and MIC	[82]
Anti-neurodegenerative	Promotor of remyelination	25 mg/kg (intraperitoneal injection)	Brain biopsies, pro-inflammatory mediators and populations, lipid peroxidation	[83]
	Attenuation of Alzheimer’s disease markers	0–20 μg/mL	Intracellular GSH, APP, BACE2, GSH-Px, SOD2, GR, BDNF, α-tubulin	[84]
		50 or 100 mg/kg (intraperitoneal injection)	Morris water maze, novel object recognition and open field test, ChAT, inflammatory and apoptotic mediators, IRS-1, INS, PI3K/AKT, and PTEN gene expression	[85]
		200 mg/kg (intraperitoneal injection)	Eight-arm radial maze, HAC3, pro-inflammatory, and proapoptic mediators	[86]
	Attenuation of Parkinson’s disease markers	2.5–7.5 μM	Fibril formation of αS or Aβ40/42, ADH, catalase	[87]
Antidiabetic	Antidiabetic and antiglycation	100–500 μg/mL	Inhibitory effect of PPA and β-glucosidase	[88]
		100 and 200 mg/kg 100 mg/kg 200 mg/kg 50 mg/kg (oral)	Blood glucose, glycosylated hemoglobin HbA1c, BUN, urea, serum creatinine, SGOT/AST, SGPT/ALT, alkaline phosphatase, total bilirubin, TGs, LDL-C, TC, and HDL-C NBT assay, carbonyl content, reduced GSH	[89,90,91,92]
Hepatoprotection	Attenuation of nephrotoxicity	Not described	Plasma urea, creatinine, urinary N-acetyl-β-D-glucosaminidase, creatinine and lithium, histomorphology evaluation	[93]
	Reduction of hepatocyte damage	50–200 mg/kg (intraperitoneal injection)	Hepatic lipid peroxidation assayed by measuring malondialdehyde	[94]
		1–100 μgM	ROS, MDA, GSH, GSH-Px, ALT, AST, SOD, TGF-β1, HGF	[71]
Nephroprotection	Prevention of cisplatin induced nephrotoxicity by reducing oxidative stress	5–30 mg/kg (intraperitoneal injection)	Blood urea nitrogen, plasma glutathione peroxidase, plasma creatinine quantification, N-acetyl-β-D-glucosaminidase, apoptosis and histopathological changes; glutathione, malondialdehyde, 4-hydroxynonenal, and oxidized proteins quantification	[95]
	Protection of Type 2 diabetes mice against oxidative stress and renal dysfunction	300 mg/kg (oral)	Urinary 8-hydroxy-2-deoxyguanosine, 8-iso-prostaglandin F_2α_ and albumin quantification; immunohistochemistry	[96]
	Recovery of cisplatin-induced renal injury in renal tissue and HK-2 cell and reduction of p-ERK, p-JNK, p-p38, Bax, caspase-9, and caspase-3	50 mg/kg (intraperitoneal injection)	Light microscopy examination, cell viability Assay, western blot, caspase-3 activity assay, and apoptosis detection by the terminal deoxynucleotidyl transferase-mediated dUTP-biotin nick end labeling method	[97]
Cardiovascular protection	Reduction of atherosclerotic disease	200 μM	HMOX-1, eNOS, P22, VCAM-1	[98]
	Prevention of AMI-induced oxidative stress, inflammation and heart damage	50 mg/kg (subcutaneous injection)	CK, AST, ALT, ROS, nitrites, oxidized glutathione, pro-inflammatory and proapoptotic cytokines, lipid peroxidation	[99]
	Prevention of cardiovascular diseases and atherosclerotic formation	0.25% and 1.25% (oral)	Cholesterol, MDA, GOT, GPT, catalase, SOD, GSH-Px, HMG CoA	[100]
Anti-obesity	Prevention of endothelial dysfunction and attenuation of obesity	2500, 5000, or 10,000 mg/kg (oral)	Serum triglyceride, total cholesterol, HDL-C, and glucose, insulin and leptin, immunohistochemistry analysis	[101]
	Reduction of adipogenesis and lipogenesis	0, 0.625, 1.25, 2.5, 5, 10, or 20 μg/mL (oral)	Western blots of adipogenic proteins (C/EBPα, PPARγ, and aP2) and lipogenic proteins (SREBP1, ACC, FAS, LPAATβ, Lipin1, and DGAT1)	[102]
Wound healing	Proliferation of fibroblasts, synthesis of ECM components and regeneration	10–200 μg/mL (superficial collagen films) 50 μg/mL	Cytotoxicity and proliferation/viability of fibroblasts, cdK1, cdK2, uPA, PI3K, and in vivo wound healing analysis	[103,104]
Antimelanogenic	Attenuation of melanin production	Not described	Cellular tyrosinase, production of melanin, DPPH	[105]
		0.05–2.00 mg/mL	Tyrosinase activity, melanin, intracellular cAMP, MITF, tyrosinase, ERK, pERK1/2, MEK1/2, p38, CREB	[106]

ABTS: 2,2′-azino-bis-(3-ethylbenzthiazoline-6-sulfonic acid), ACC: acetyl-CoA carboxylase, ALT: alanine transaminase, APP: amyloid precursor protein, aP2: adipocyte protein 2, ALT: alanine transaminase, AST: aspartate transaminase, BACE: β-site APP-cleaving enzyme, BALF: bronchoalveolar lavage fluid, BDNF: brain-derived neurotrophic factor, BUN: blood urea nitrogen, cAMP: Cyclic adenosine monophosphate, cdK: cyclin-dependent kinases, C/EBPα: CCAAT/enhancer-binding protein alpha, ChAT: choline acetyltransferase, CK: creatine kinase, COX: cyclooxygenase, CREB: cAMP-response element binding protein, DCF: 2,7-dichlorofluorescein, DCFDA: 2′,7′-dichlorodihydrofluorescein diacetate, DGAT: diacylglycerol acyltransferases, DPPH: α, α, diphenyl-β-picrylhydrazyl, ECM: extracellular matrix, EIA: enzyme immunoassay, ELISA: enzyme-linked immunosorbent assay, EMT: epithelial-mesenchymal transition, eNOS: endothelial nitric oxide synthase, ERK: extracellular signal-regulated protein kinases, FAS: fatty acid synthase, FRAP: ferric reducing antioxidant power, GOT: glutamate-oxaloacetate transaminase, GPT: glutamate-pyruvate transaminase, GR: glutathione reductase, GSH: glutathione, GSH-Px: glutathione peroxidase, HDAC: histone deacetylase, HDL-C: high-density lipoprotein cholesterol, HGF: hepatocyte growth factor, HIF-1α: hypoxia-inducible factor 1-alpha, HMG CoA: 3-hydroxy-3-methylglutaryl-coenzyme A, HMOX: heme oxygenase, IL: interleukin, iNOS: inducible nitric oxide synthase, INS: insulin, IRS: insulin receptor substrate, LDH: lactate dehydrogenase, LDL-C: low-density lipoprotein cholesterol, LPAAT: lysophosphatidic acid acyltransferase, MBC: minimum bactericidal concentration, MDA: malondialdehyde, MIC: minimal inhibitory concentration, MITF: microphthalmia-associated transcription factor, MMP: matrix metalloproteinase, MPO: myeloperoxidase, NBT: nitro blue tetrazolium, NF-kB: nuclear factor-kappa B, NQO: NAD(P)H quinone oxidoreductase, PGE2: Prostaglandin E₂, PI3K/AKT: phosphatidylinositol-3-kinase/serine threonine protein kinase B, PPA: porcine pancreatic amylase, PPARγ; peroxisome proliferator-activated receptor gamma, PTEN: phosphatase and tensin homolog, qRT-PCR quantitative real-time-polymerase chain reaction, ROS: reactive oxygen species, SGOT/AST: serum glutamic oxaloacetic transaminase/aspartate aminotransferase, SGPT/ALT: serum glutamic pyruvate transaminase/alanine aminotransferase, SOD: superoxide dismutase, SMA: smooth muscle actin, SREBP: sterol regulatory element-binding protein, TC: total cholesterol, TEM: transmission electron microscopy, TIMP: tissue inhibitor matrix metalloproteinase, TNF: tumor necrosis factor, TGs: triglycerides, TGF-β1: transforming growth factor-beta1, uPA: urokinase-type plasminogen activator, VCAM: vascular cell adhesion molecule, VEGF-A: vascular endothelial growth factor A, WB: western blot, 7-AAD: 7-amino-actinomycin D.

**Table 3 pharmaceuticals-16-00592-t003:** Description of the extraction methods used for disrupting the phycocyanin cell membrane, their conditions, advantages, and limitations.

Cell Disruption Method	Purity *	Yield (mg/g)	Conditions	Advantages	Limitations	Ref.
Freeze/thaw cycles	0.56–0.66-	73.73–74.51	6 cycles, −40 °C/4 h + room temperature/1 h, 0.1 M phosphate buffer pH (6.8), 1:6, 1:8 and 1:10 S/L ratios	Continuous damage to the plasmatic membrane, easy to perform, availability	Time and energy consuming, and often require high solvent, leading to an increase in the production costs. Not suitable for industrial scale	[141]
	0.4	ND	25 °C/4 h, distilled water, 1:25 S/L ratio			[142]
	0.77	217.18	3 cycles, −20 °C + room temperature/24 h, 20 mM of sodium acetate and 50 mM of NaCl buffer (pH 5.1) 1:20 S/L ratio			[143]
	2.10	41.90	4 cycles, −20 °C/4 h + room temperature/1.5 h, sodium hydroxide, (pH 6.8), 1:20 S/L ratio			[144]
Mixing and homogenization	0.6	52.11	25,200× *g*/10 min, 0.1 M phosphate buffer (pH 6.8), 1:6 S/L ratio	Simple, availability, reproducibility	Increased temperatures during the process, time consuming, not suitable for industrial scale, cell debris released	[141]
	ND	ND	Rotary shaker at 30 °C, 10 mM sodium phosphate buffer (pH 7.0), 10 mM sodium acetate buffer (pH 5.0), NaCl 0.15 M and CaCl2 10 g/L, 1:25 S/L ratio			[142]
	ND	67.61	1.71% biomass/solvent ratio, 6237.66 homogenization rate, 15 min extraction time			[145]
	0.67	103.07	Oven-dried biomass preparation, 70 °C/4 h, extraction at 25 °C/24 h by 0.01 M phosphate buffer using homogenization assisted method at 0.02 g/mL biomass concentration			[146]
Bead milling	0.46	217.14	Low-density beads for low viscosity media, 80–85% degree of bead filling	Low time, high biomass disruption, low energy	Time very dependent of the type of bead, low purities, cell debris, additional purification steps needed	[144]
	ND	119.48 mg/g	Bead diameter 0.3 mm, glass beads at a speed of 3580 rpm. 4 cycles of milling each 25 s and subsequent cooling at 4 °C			[147]
	ND	90% recovery	Dakot Zirconia beads (0.5–1.4 mm), 330 rpm agitation, 8 h under 0.5 M Ca (II) in a 0.35 M acetate buffer (pH 6.8)			[148]
	0.21	ND	Bead Beater, diameter 0.1 mm, agitation of 4800 rpm, 10 cycles of 10 s. Following each cycle, samples cool down in water at 0 °C			[149]
	ND	94.90	Glass beads 0.25–0.5 mm of diameter) in 2 mL flasks, 4 cycles of 25 s at 30 Hz of vibrational frequency			[150]
Ultrasound	0.62	51.51	Pre-soaked for 120 min, ultrasonication amplitude of 50%, 2.5 min, 1:6 S/L	High purities, reproducibility, suitable for industrial scale, temperature could be controlled	Increased temperatures during the process, complex process, expensive, specific equipment	[141]
	ND	98.84	1% biomass/solvent ratio, 60% amplitude, 16.23 min extraction time			[145]
	0.67–0.93	90.00	Frequencies of 20–100 kHz, power intensities >1 W/cm^2^, PBS soaking			[146]
	0.65	18.20	Power 60 W, extraction for 10 s, 30 cycles in total, in ice bath			[151]
Electric field	2.45	143.33	Freeze/thaw and pulsed electric field maximum charging voltage of 30 kV, square bipolar pulses with a variable pulse width of 4–32 μs and a pulse frequency up to 300 Hz.	Increased permeability of the membrane	Long time to optimization, complex equipment, intracellular compounds might not be completely released	[141]
	0.51	151.94	40 °C, 25 kV/cm, 150 μs			[149]
	ND	ND	50 to 200 pulses at 20 kV			[152]
High-pressure homogenization	ND	ND	3.5 min with pressures between 50 and 600 MPa, distilled water ratio of 6% (wt%)	Simpler, scalable for industrial application, environmentally friendly, high recovery	Expensive, not useful in extracting dry biomass, can lead to protein denaturation	[152]
	ND	291.90	100 mM Na-phosphate solvent (pH 7), 1400 bar			[153]
	1.2–1.4	90% recovery	300 MPa for 10 min, deionized water or phosphate buffer (pH 6.8), 1/20 (*w*/*v*) ratio			[154]
Enzyme-assisted	0.80–0.90	20–25	1 mg/mL lysozyme, high pressure homogenizer D-15M at 10–12,000 p.s.i., 4–8 °C	Stable, efficient, eco-friendly	More efficient when combined with other methods	[153]
	1.19	82.07	1.0% enzyme concentration, 16 h incubation time, 1:6 S/L ratio			[155]
	1.09	92.73	2.5 min Ultrasonication at 50% Amplitude, 0.6% enzyme concentration, 16 h incubation, 1:6 S/L ratio			

* Phycocyanin purity index: A620/A280 > 0.7—food grade, >1.5—cosmetic grade, >3.9—reagent grade, >4.0—analytical grade [41]. ND: Not determined.

**Table 4 pharmaceuticals-16-00592-t004:** Description of the methods of purification used to improve the purity of phycocyanin recovered from Spirulina, the conditions used, and the values of purity and recovery obtained by each method.

Purification Method	Conditions	Purity *	Recovery (%)	Ref.
Ammonium sulfate (AS) precipitation	50–65% AS	2.11	86	[197]
Chitosan and activated charcoal	0.24% chitosan/8.4% activated charcoal	3.14	79	[198]
Stirred fluidized bed IEC	1%, *dw*/*v* in STREAMLINE DEAE 10%, *dw*/*v* in STREAMLINE DEAE	2.70 3.00	90 64	[199]
IEC	IEC with pH gradient using an anion-exchanger Q-Sepharose Fast Flow column	4.20	49	[200]
Combined methods	50–65% AS, dialysis in sodium acetate buffer, IEC on a DEAE-Sepharose Fast Flow column	5.59	67	[197]
	50% AS, dialysis, ultrafiltration with MWCO of 50 kDa and IEC (anion-exchanger resin Q-Sepharose Fast Flow column) with pH gradient	4.00	80	[44]
	65% AS, 65% dialysis with 12–14 kDa membranes, microfiltration, IEC in a Sephadex-G-100 column, and HPLC with a reverse column	92%	53	[88]
	1.113 M AS, filtration with a PVDF membrane and two hydrophobic interaction membrane chromatography steps	4.20	67	[201]
	2% *w*/*v* chitosan solution, 80 g/L activated charcoal, ultrafiltration, and IEC on DEAE Sephadex A-25	4.30	42	[151]

AS: ammonium sulfate, IEC: ion exchange chromatography, DEAE: diethylaminoethyl, HPLC: high performance liquid chromatography, MWCO: molecular weight cut-off, PVDF: polyvinylidene fluoride. * Phycocyanin purity index: A615-620/A280 > 0.7—food grade; >1.5—cosmetic grade, >3.9—reagent grade, >4.0—analytical grade [41].

## Data Availability

Not applicable.

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
