# Peer review of "Exploring the Benefits of Phycocyanin: From Spirulina Cultivation to Its Widespread Applications"

_pharmaceuticals, 2023, doi:10.3390/ph16040592_

Round 1
Reviewer 1 Report
It is a review article about some phycocyanin extraction and purification methods from Spirulina algae. In addition, the authors describe its main generalities and mention some beneficial properties that this phycocyanobilin possesses.
The following details were found in the manuscript:
Some errors were found in the degree symbols used throughout the text, as well as the units in lines 574, 577-578. Besides, the symbols of some ROS are not correct.
It is required to verify the colors described in the caption of figure 2.
Verify the references with what is mentioned in the text, since experimental data are described (from the original paper), and review articles are referenced (Table 2).
Phycocyanin possesses several properties that are not mentioned in Table 2 or in the text.
Lines 278-279. How much was consumed, and in which model was demonstrated the increase in SOD enzyme? Why did this increase exist?
In the antioxidant section, more information that phycocyanin possesses as an antioxidant is also missing. The same for its anticancer and antibacterial property. I suggest that the antibacterial properties of phycocyanin should be mentioned in a table.
It would be essential to know the concentrations, model and mechanism of the properties mentioned throughout the review, since only the property is mentioned in some sections, but not the mechanism.
Review existing clinical studies of phycocyanin.
Images illustrating part of the content of the review are required.
Section 3.4.6. It would be essential to mention the range of purity obtained
What would be the best method, process and conditions for extracting and purifying phycocyanin?
Will it depend on the quality of the seaweed, its dehydration, etc.?
Although section 3.2 mentions some properties of phycocyanin, there are many more, in this sense it is a summary of its properties that it would be enough just to mention them but not make subsections of these properties where they are described superficially. I believe that section 3.3 should be further strengthened and provide more figures that illustrate these processes.
Author Response
Dear Editor of Pharmaceuticals,
In reply to the review performed on the paper entitled “From Spirulina to the world: unveiling the excellence of phycocyanin”, we would like to acknowledge the valuable comments performed by the editor that kindly accepted to revise our manuscript. We would like to confirm that we have addressed most issues and answered the questions made by reviewers. We hope the answers below and modifications that have been done in the manuscript are clear and concise enough as required by the reviewers to enable the publication of the manuscript in Pharmaceuticals.
Answer to referee’s comments and queries
Detailed responses to Reviewer 1
Reviewer´s comment: Some errors were found in the degree symbols used throughout the text, as well as the units in lines 574, 577-578. Besides, the symbols of some ROS are not correct.
Our reply: Thank you for the suggestion. The symbols were properly corrected (lines 268-279).
Reviewer´s comment: It is required to verify the colors described in the caption of figure 2.
Our reply: Thank you for the comment. In fact, colors of figure 2)B and C) are represented according the figure mentioned in Biao Yuan et al. report (DOI: 10.1016/j.crfs.2022.11.019) and as represented in RCSB Protein data bank (https://www.rcsb.org/structure/1GH0), where alpha subunit is represented by green, orange, and pink and the beta subunit by grey, yellow, and brown colors.
Reviewer´s comment: Verify the references with what is mentioned in the text, since experimental data are described (from the original paper), and review articles are referenced (Table 2).
Our reply: Thank you for the comment. The references mentioned in the text and Table 2 were revised and substituted according to the original papers.
Reviewer´s comment: Phycocyanin possesses several properties that are not mentioned in Table 2 or in the text.
Our reply: Thank you for the comment. In fact, the biological properties of phycocyanin depends on the algae source which is extracted. In addition, to the biological functions that are highlighted in Table 2 and across the text. Nevertheless, as suggested, it was mention that Table 2 describes the main biofunctions of phycocyanin from Spirulina described in the literature (lines 233-234).
Reviewer´s comment: Lines 278-279. How much was consumed, and in which model was demonstrated the increase in SOD enzyme? Why did this increase exist?
Our reply: Thank you for the question. The information was added (lines 286-291).
Reviewer´s comment: In the antioxidant section, more information that phycocyanin possesses as an antioxidant is also missing. The same for its anticancer and antibacterial property. I suggest that the antibacterial properties of phycocyanin should be mentioned in a table.
Our reply: Thank you for the suggestion. More detailed information was added in these sections regarding the antioxidant and anticancer properties (lines 268-375). Concerning the antimicrobial activity of phycocyanin recovered from Spirulina, the available information was already added in the Table 2. In fact, there are more studies that demonstrate antimicrobial activity but refer to phycocyanin obtained from other microalgae, which is not the focus of this review.
Reviewer´s comment: It would be essential to know the concentrations, model and mechanism of the properties mentioned throughout the review, since only the property is mentioned in some sections, but not the mechanism.
Our reply: Thank you for the suggestion. The information was added in the appropriate sections.
Reviewer´s comment: Review existing clinical studies of phycocyanin
Our reply: Thank you for the suggestion, it was already added (lines 480-485)
Reviewer´s comment: Images illustrating part of the content of the review are required.
Our reply: Thank you for the suggestion, but we consider that putting images for each method that was described would make the document too long. Thus, so it was added one schematic figure, Figure 3 (Line 495), to properly introduce the sections 3.3, 3.4 and 3.5. Also, all information has been revised and added across the text and tables.
Reviewer´s comment: Section 3.4.6. It would be essential to mention the range of purity obtained
Our reply: Thank you for the suggestion. The information required was added (when available in the original articles) in the Section 3.4.6 text and Table 3.
Reviewer´s comment: What would be the best method, process and conditions for extracting and purifying phycocyanin?
Our reply: Thank you for the pertinent question. As described in the manuscript, several methods of extraction and purification of phycocyanin from Spirulina have been reported. However, the best method and respective conditions will depend on several factors, considering the time, cost and yield of the processes. The intended amount of phycocyanin, small- or large-scale production and the final application of the product are determining factors in choosing the methods to be used.
Given that this information was not clearly explicit in the manuscript, it was added to the text (Lines 489-494).
Reviewer´s comment: Will it depend on the quality of the seaweed, its dehydration, etc.?
Our reply: Thank you for the question. The focus of this review is the phycocyanin extracted from Spirulina. As already answered in the previous question, the best method and respective conditions depend on several factors, including the growth phase of the algae, permeability of the membrane that varies depending on the algae from which phycocyanin is extracted. Also, depending on the purpose of phycocyanin, several conditions of the methods, particularly temperature, pH and light, stability should be optimized to ensure phycocyanin stability and prevent its degradation. This information is already highlighted throughout the topic 3.4 (Key parameters for phycocyanin extraction and stability)
Reviewer´s comment: Although section 3.2 mentions some properties of phycocyanin, there are many more, in this sense it is a summary of its properties that it would be enough just to mention them but not make subsections of these properties where they are described superficially. I believe that section 3.3 should be further strengthened and provide more figures that illustrate these processes.
Our reply: Thank you for the suggestion. Regarding section 3.2, the first part has been considered as indicated previously. The suggestion regarding the subsections on the biofunctions of phycocyanin was accepted. Regarding section 3.3, information was added, but as mentioned previously, we consider that putting images for each method that was described would make the document too long, so it was added one figure, Figure 3, to summarize the topics properly schematically 3.3, 3.4 and 3.5. Moreover, the text has been revised and the missing critical information (when available) from the original articles was added as suggested.
Sincerely,
Ana Isabel Ramos Novo Amorim de Barros

Reviewer 2 Report
The reviewed paper on phycocyanin from Spirulina is an extensive and comprehensive review of the literature. The manuscript is well laid out and presents the information gathered well. I have no objections to the substantive content, but the authors did not avoid some errors, rather technical ones, which I ask for correction.
- line 140 - the word "industry" is probably missing after listing the directions of application
- Figure 2 - The pattern shown is actually the pattern of the phycocyanobilin chromophore itself, please clarify the caption
- lines 270 - 272 - please correct the notation of the ROS formulas: superoxide anion and peroynitrite should have "-" sign in superscript, please correct the subscripts in the formulas (O2- and H2O2)
- line 276 - DPPH's most common chemical name is 1,1-Diphenyl-2-picrylhydrazyl not using Greek letters
- when writing Latin names of bacteria, it is not necessary to provide abbreviated names - enter the full name for the first time, then use the abbreviated name, it is generally accepted (e.g. first time Escherichia coli, then E. coli). In the case of generic names with "sp", this part of the name should not be italicized
- line 350 - correct the format of Salmonella enteritidis name
-lines 574-578 - please correct the units - "m" should not be superscript and "s" should be lower case
After these changes, I am applying for acceptance of the manuscript.
Author Response
Dear Editor of Pharmaceuticals,
In reply to the review performed on the paper entitled “From Spirulina to the world: unveiling the excellence of phycocyanin”, we would like to acknowledge the valuable comments performed by the editor that kindly accepted to revise our manuscript. We would like to confirm that we have addressed most issues and answered the questions made by reviewers. We hope the answers below and modifications that have been done in the manuscript are clear and concise enough as required by the reviewers to enable the publication of the manuscript in Pharmaceuticals.
Detailed responses to Reviewer 2
Reviewer´s comment: line 140 - the word "industry" is probably missing after listing the directions of application.
Our reply: Thank you for the suggestion that was already corrected (line 141).
Reviewer´s comment: Figure 2 - The pattern shown is the pattern of the phycocyanobilin chromophore itself, please clarify the caption.
Our reply: Thank you for the comment. It was considered in lines 212-213. In fact, Figure 2C) represents one of the phycocyanobilin chromophores present in phycocyanin subunits.
Reviewer´s comment: lines 270 - 272 - please correct the notation of the ROS formulas: superoxide anion and peroynitrite should have "-" sign in superscript, please correct the subscripts in the formulas (O2- and H2O2).
Our reply: Thank you for the suggestion and explanation. The subscripts were properly corrected (lines 268-279).
Reviewer´s comment: line 276 - DPPH's most common chemical name is 1,1-Diphenyl-2-picrylhydrazyl not using Greek letters.
Our reply: Thank you for the explanation, the chemical name was corrected as suggested.
Reviewer´s comment: when writing Latin names of bacteria, it is not necessary to provide abbreviated names - enter the full name for the first time, then use the abbreviated name, it is generally accepted (e.g. first time Escherichia coli, then E. coli). In the case of generic names with "sp", this part of the name should not be italicized.
Our reply: Thank you for the suggestion and explanation that was already corrected in lines 386-404.
Reviewer´s comment: line 350 - correct the format of Salmonella enteritidis name
Our reply: Thank you for the suggestion but Salmonella Enteritidis is the correct format to mention the Enteritidis serotype of Salmonella enterica.
Reviewer´s comment: lines 574-578 - please correct the units - "m" should not be superscript and "s" should be lower case.
Our reply: Thank you very much for the explanation, the units were corrected as required (lines 636-640).
Sincerely,
Ana Isabel Ramos Novo Amorim de Barros

Round 2
Reviewer 1 Report
This review is about the properties of phycocyanin and the methods of extraction and purification of phycocyanin from Spirulina algae.
The details I found are:
This review aims to update the properties and purification methods of Spirulina phycocyanin. However, this is not reflected in the title.
It would be necessary for the authors also give a critique and not just describe the information.
Check the references in Table 2. Now other authors are cited, unlike V1 of the manuscript. Or, update that table.
Table 2. Only some of the main functions of phycocyanin are mentioned, but other functions that are also important such as renal, hepatic, etc., are still not mentioned.
The authors did not respond to the increase in SOD by phycocyanin.
Despite the length of the review, I think images are needed. The new image is tiny. In this image, it would be necessary to place the name of some figures that cannot be easily interpreted due to their small size (light bulb, thermometer, and others).
The degree symbol (°) still appears incorrectly as º.
In the conclusions, the authors mention that other Spirulina components could be used in the industry. How which ones?
Author Response
Dear Editor of Pharmaceuticals,
In reply to the review performed on the paper entitled “From Spirulina to the world: unveiling the excellence of phycocyanin”, we would like to acknowledge the valuable comments performed by the editor that kindly accepted to revise our manuscript. We would like to confirm that we have addressed most issues and answered the questions made by reviewer 1. We hope the answers below and modifications that have been done in the manuscript are clear and concise enough as required by the reviewer to enable the publication of the manuscript in Pharmaceuticals.
Answer to referee’s comments and queries
Detailed responses to Reviewer 1
Reviewer´s comment: This review aims to update the properties and purification methods of Spirulina phycocyanin. However, this is not reflected in the title.
Our reply: Thank you for the suggestion. The title changed to: “Exploring the Benefits of Phycocyanin: from Spirulina cultivation to its widespread applications”
Reviewer´s comment: It would be necessary for the authors also give a critique and not just describe the information.
Our reply: The authors are grateful for the reviewer's comments, however, the primary objective of our review is to provide a comprehensive summary of the existing literature and not to provide a critical evaluation of the information presented. We think this is the most correct way to approach a review article.
Reviewer´s comment: Check the references in Table 2. Now other authors are cited, unlike V1 of the manuscript. Or, update that table.
Our reply: Thank you for the comment. The other authors/references introduced in table 2 are now highlighted.
Reviewer´s comment: Table 2. Only some of the main functions of phycocyanin are mentioned, but other functions that are also important such as renal, hepatic, etc., are still not mentioned.
Our reply: Thank you for the comment. More information was added in section 3.2 (lines 468-505) and table 3.
Reviewer´s comment: The authors did not respond to the increase in SOD by phycocyanin. (How much was consumed, and in which model was demonstrated the increase in SOD enzyme? Why did this increase exist?)
Our reply: Thank you for the question. The information was already added in lines 299-318.
Reviewer´s comment: Despite the length of the review, I think images are needed. The new image is tiny. In this image, it would be necessary to place the name of some figures that cannot be easily interpreted due to their small size (light bulb, thermometer, and others).
Our reply: Thank you for the suggestion. In Table 3 (now Figure 4), a label showing the symbols has been added to facilitate their interpretation because of their small size. Figures 3 and 5, which reflect the biofunctions of phycocyanin and its purification methods, respectively, were added.
Reviewer´s comment: The degree symbol (°) still appears incorrectly as º.
Our reply: Thank you for the suggestion. The symbol was corrected across the text and tables.
Reviewer´s comment: In the conclusions, the authors mention that other Spirulina components could be used in the industry. How which ones?
Our reply: Thank you for the question. More specific information was added (lines 185-195).
Sincerely,
Ana Isabel Ramos Novo Amorim de Barros

Round 3
Reviewer 1 Report
This review is about the properties of phycocyanin and the methods of extraction and purification of phycocyanin from Spirulina algae. In this version I have found some details which are described below
· Although Figures 4 and 5 are different, they are very similar. For this reason, I suggest that Figure 4 should be simplified (but not the exact figure as the one in Figure 5).
· Please check the abbreviations, as CCl4 is incorrect (line 471).
· In Table 2, it would be convenient to modify the concentration per dose in the in vivo models and add the route of administration.
· Review the references since they are repeated.
· I think the review has improved a lot. Greetings.
Author Response
Dear Editor of Pharmaceuticals,
In reply to the review performed on the paper entitled “Exploring the Benefits of Phycocyanin: from Spirulina cultivation to its widespread applications", we would like to acknowledge the valuable comments performed by the editor that kindly accepted to revise our manuscript. We would like to confirm that we have addressed most issues and answered the questions made by reviewer 1. We hope the answers below and modifications that have been done in the manuscript are clear and concise enough as required by the reviewer to enable the publication of the manuscript in Pharmaceuticals.
Answer to referee’s comments and queries
Detailed responses to Reviewer 1
Reviewer´s comment: Although Figures 4 and 5 are different, they are very similar. For this reason, I suggest that Figure 4 should be simplified (but not the exact figure as the one in Figure 5).
Our reply: Thank you for the comment. The Figures 4 and 5 were changed as suggested.
Reviewer´s comment: Please check the abbreviations, as CCl4 is incorrect (line 471).
Our reply: Thank you for the comment that was already corrected (lines 471 and 474).
Reviewer´s comment: In Table 2, it would be convenient to modify the concentration per dose in the in vivo models and add the route of administration.
Our reply: Thank you for the suggestion. All available information from each study, concentration or dose, and route of administration, were added in Table 2, including for the in vivo assays.
To better clarify, it was altered the title of the 3rd column of Table 2 to “Dose or concentration (route of administration)”.
Reviewer´s comment: Review the references since they are repeated.
Our reply: Thank you for the comment. In the last version of the manuscript reference 57=129, reference 58=107, reference 69=82, reference 118=122. We updated the references, and the repeated ones were eliminated.
Sincerely,
Ana Isabel Ramos Novo Amorim de Barros
